# Mesoscale simulation of biomembranes with FreeDTS

Weria Pezeshkian [1] ✉ & John H. Ipsen[2]

We present FreeDTS software for performing computational research on biomembranes at the mesoscale. In this software, a membrane is represented by a dynamically triangulated surface equipped with vertex-based inclusions to integrate the effects of integral and peripheral membrane proteins. Several algorithms are included in the software to simulate complex membranes at different conditions such as framed membranes with constant tension, vesicles and high-genus membranes with various fixed volumes or constant pressure differences and applying external forces to membrane regions. Furthermore, the software allows the user to turn off the shape evolution of the membrane and focus solely on the organization of proteins. As a result, we can take realistic membrane shapes obtained from, for example, cryo-electron tomography and backmap them into a finer simulation model. In addition to many biomembrane applications, this software brings us a step closer to simulating realistic biomembranes with molecular resolution. Here we provide several interesting showcases of the power of the software but leave a wide range of potential applications for interested users.

Fluid artificial and biological membranes can adapt to a diverse range of morphologies from multi-spherical structures to the astonishing forms seen in the subcellular organelles such as endoplasmic reticulum, Golgi apparatus, and mitochondria[1,2]. These shapes are often dynamic and constantly undergo significant changes that are crucial for cell function e.g., such as endocytosis, cellular respiration. Also, biological membrane shapes contain information about overall health of an organism, the physiological state of the cell and abnormal membrane architectures are implicated in many diseases such as Parkinson's Disease[3–7]. The shape classes of simple lipid membranes are well understood and they can be described by a few macroscopic parameters such as spontaneous membrane curvature and pressure difference[1]. The membranes of cells, however, are much more complex, and distinct mechanisms govern their overall arrangement at various scales, ranging from the molecular to the macroscale[8–10]. Thus, the study of membrane organization continues to be an active and essential field of science. Computer simulations are an effective tool for studying biomembrane architecture and the mechanisms involved in its remodeling[11]. For analyzing membrane shape at length scales of up to 100 nm, molecular simulations, such as molecular dynamics and dissipative particle dynamics simulations, have been very effective. Examples of exciting developments include the local curvature of membranes caused by a single protein or by the assembly of proteins[12–14], membrane shape-induced lipid sorting[15], wetting-induced membrane deformation[16], and even curvature sensing of proteins[17]. Due to limitations in accessibility of large time and length scales, molecular-based simulations alone cannot provide a comprehensive picture of membrane remodeling. On the other end, macroscopic modeling that incorporates protein effects as a mean field density function enables the description of large-scale and generic membrane remodeling behaviors. However, it overlooks many phenomena which are the result of membrane fluctuations and rotational and translational entropy of single and (few) proteins[8,18,19]. In between these two scales, mesoscopic modes in which large biomolecules such as proteins are explicitly considered, while lipids are modeled in a mean field manner (but allows for undulations and shape fluctuations) are needed to fully map out the organization of complex membranes[20,21].

[1]Niels Bohr International Academy, Niels Bohr Institute, University of Copenhagen, Blegdamsvej 17, 2100 Copenhagen, Denmark. [2]MEMPHYS/PhyLife, Department of Physics, Chemistry and Pharmacy (FKF), University of Southern Denmark, Campusvej 55, 5230 Odense M, Denmark. ✉e-mail: weria.pezeshkian@nbi.ku.dk

Several mesoscopic models have been used to explore diverse range of membrane-associated processes such membrane shape remolding by BAR-domain proteins and crowding of intrinsically disordered proteins[22–24], as protein clustering via membrane-mediated interaction[25], membrane neck constriction by assembly of proteins[26] and even activity-driven membrane remodeling[27] (for more see ref. 20 and the reference within). In spite of this, these studies are often conducted with in-house software, or/and the software is limited to those specific applications that are difficult to apply to new research questions, which has hampered progress in mesoscopic membrane modeling. Recently, a software package called TriMem has been released that facilitates the simulation of triangulated surfaces with a strong focus on performance, in order to integrate the evolution of a system efficiently[28]. However, this package currently lacks the protein model and only applicable to simple lipid bilayers.

Here, we present FreeDTS, a software package for the computational investigation of biomembranes, at the mesoscopic length scale which can also be used for macroscale modeling of membranes. We have chosen the name FreeDTS because it is free to use and free from any external library apart from the C++ Standard Library. DTS refers to Dynamically Triangulated Surfaces. The software is designed to cover a diverse range of biological processes and is easy to expand to cover more. In FreeDTS, a membrane is represented by a dynamically triangulated surface equipped with vertex-based inclusions to integrate the effects of integral and peripheral membrane proteins. Several algorithms are implemented in FreeDTS to perform complex membrane simulations in different conditions, e.g., framed membranes with constant tension. Membranes also can be confined into a fixed region of the space to explore the effect of the environment on the membrane shape and fluctuations. In addition, FreeDTS allows one to turn off the shape evolution of the membrane and only explore organization of membrane proteins. This allows us to take realistic membrane shapes obtained, for instance, from cryo-electron tomography and obtain heterogeneous organization of biomolecules which can be backmapped to finer simulation models. This feature with helps from backmapping schemes e.g., TS2CG, brings us a step closer to simulating realistic biomembranes with molecular resolution[29,30]. In the following, we report several interesting examples to show the power of the software and the detailed information of how to use the software is included in the manual.

## Results

In this section, we present a number of results to demonstrate the power of FreeDTS in capturing membrane mesoscale organization. These include capturing membrane undulation spectrum, protein-induced membrane deformation, tether pulling, curvature-induced protein sorting, and finally protein sorting by real mitochondrial membrane shape (the proteins are hypothetical and do not represent any realistic mitochondrial proteins).

### Undulation of framed membranes

Flat membranes are a very common and pragmatic model of membrane segments as the curvature of typical cellular or model membranes is small and can be locally considered flat. FreeDTS allows simulations of flat membranes within a periodic box with constant frame tension. Theoretical analysis, confirmed by experiment and molecular simulations shows that a free membrane undulation spectrum (with the exception of membranes with small bending rigidities[31]) follows Eq. (1)[32–35],

$$\langle u(q)u(-q) \rangle = \frac{1}{\kappa_{\text{eff}} q^4 + \tau q^2} \tag{1}$$

where $\kappa_{\text{eff}}$ and $\tau$ are effective bending rigidity (renormalized bending rigidity) and frame tension respectively (Supplementary Fig. 1). Accordingly, undulation spectrum for membranes under tension; follows $q^{-4}$ for large $q$ (small wavelength) and $q^{-2}$ for small $q$, while for tensionless membranes it follows $q^{-4}$. (see Fig. 1A). It is, however, more difficult to obtain the spectrum for membranes in different conditions, such as complex membranes or membranes in confinement. Using FreeDTS, we are able to obtain membrane undulation spectrums under a variety of different conditions. Figure 1B shows undulation spectrums for three different flat membranes (1) framed membrane with zero tension and 20% coverage by inclusions (2,3) framed membrane with zero tension confined between two sandwiching walls for two different values of wall-to-wall distance.

### Membrane shape deformation by proteins

Proteins are one of the main drivers of membrane deformations[36,37]. FreeDTS allows for exploring the membrane-shaping and sorting capacity of membrane proteins. Figure 2 shows several examples of membrane deformation by different membrane proteins at the

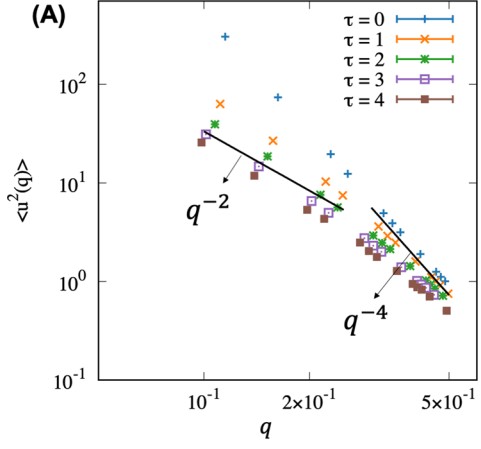

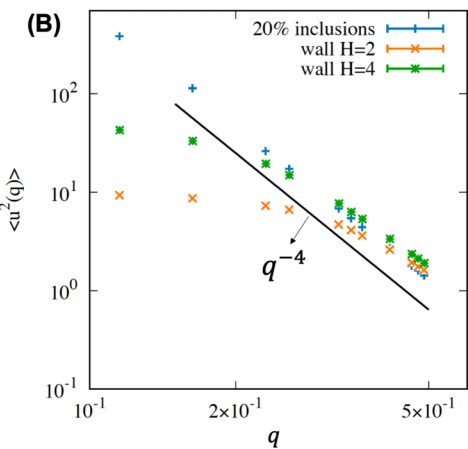

**Fig. 1 | Undulations spectrum for different membranes obtained using FreeDTS. A** Undulation spectrums for membranes with five different values of frame-tensions ($\tau$); for large $q$ (small wavelength) it follows $q^{-4}$, while for small $q$, it follows $q^{-2}$. **B** undulations spectrum for membranes in different conditions; (blue) framed membrane with zero tension and 20% inclusion coverage (orange) membranes between two sandwiching walls with a wall-to-wall distance of $H = 2l_{dts}$,

(green) membranes between two sandwiching walls with a wall-to-wall distance of $H = 4l_{dts}$. The data are obtained by simulating ten different replicas, each for 10 million Monte Carlo steps for every system. The initial one million Monte Carlo steps were disregarded in the analysis. Note, the unit of frame-tensions is $k_B T/l_{dts}^2$ in FreeDTS. Source data are provided as a Source Data file.

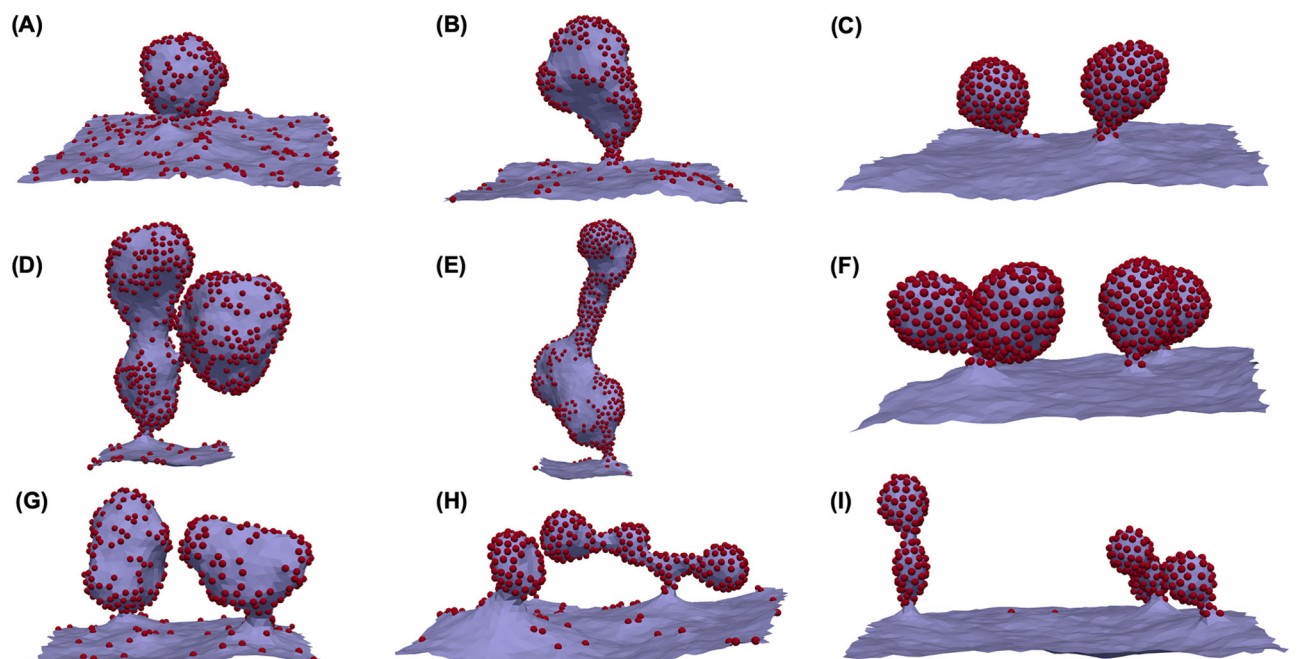

**Fig. 2 | Protein-induced membrane deformation.** Snapshots depicting the final state of membrane simulations under periodic boundary conditions following 10 million Monte Carlo steps. The red spheres are the proteins. **A**−**C** In total, 20% protein coverage, protein type one with $\Delta\kappa = 0$, $c_0 = 0.4[l_{dts}^{-1}]$, $\Delta\kappa_G = 0$ and **A** $e_{inc,inc} = -0.5[k_BT]$, **B** $e_{inc,inc} = -1[k_BT]$, **C** $e_{inc,inc} = -2[k_BT]$; **D**−**F** 40% protein coverage, protein type one with $\Delta\kappa = \Delta\kappa_G = 0$, $c_0 = 0.4[l_{dts}^{-1}]$, and **D** $e_{inc,inc} = -0.5[k_BT]$; **E** $e_{inc,inc} = -1[k_BT]$; **F** $e_{inc,inc} = -2[k_BT]$; **G**−**I** 20% protein coverage, protein type one with $\Delta\kappa = 0$, $c_0 = 0.6[l_{dts}^{-1}]$ $\Delta\kappa_G = 0$ and **G** $e_{inc,inc} = -0.5[k_BT]$; **H** $e_{inc,inc} = -1[k_BT]$; **I** $e_{inc,inc} = -2[k_BT]$.

mesoscale. For small protein–protein interactions (above a certain $c_0$ threshold[38]) buds form, and the membrane is divided into two domains of protein-rich and protein-poor. However, for larger protein–protein interactions, proteins cluster and form buds, leaving the insignificant number of proteins on the flat surface. The non-zero protein–protein interactions lead to a line tension effect at the boundary that can energetically assist the budding process, especially for large protein–protein interactions[39]. However, budding may occur without a line tension effect solely as a result of the high concentration of inclusions that induce membrane curvature[38].

While these results can describe a wide range of processes involving remodeling of membrane shapes by proteins, one can convert $l_{dts}$ to a physical unit when for example a specific protein is under consideration. For instance, if we consider that our proteins are B subunit of cholera or Shiga toxins (they have a similar lateral size of $\sim 7.2$ nm[40,41]), then $l_{dts} \sim 6.9$ nm (see Supplementary Note 1). Therefore, $c_0 = 0.4\,l^{-1} \sim 0.058$ nm$^{-1}$. This is actually very close to the reported curvature induced by these proteins ($\sim 0.056$ nm$^{-1}$ for cholera toxin and $\sim 0.07$ nm$^{-1}$ for Shiga toxin). Also, in this case, the total surface area of the membranes will be $\sim 0.15\,\mu$m$^2$. For Shiga toxin, protein–protein interactions are primarily driven by close-distance membrane fluctuation-induced forces $\sim 1k_BT$, which cannot be captured by this model and must be included directly[42]. However, it is still unknown what causes the clustering of cholera toxin. Therefore, the results of the first and second columns of Fig. 2 are expected for the B subunit of Shiga toxin which is also very similar to the shape reported in experimental settings[43]. In contrast, all configurations are possible for cholera toxin, depending on the range of its protein–protein interactions.

### Pulling a membrane tether

One of the common procedures to deform membranes is pulling a tether (nanotube) using, for example, optical tweezers[12,44,45]. Since FreeDTS allows for membrane simulations with constant tension, it can also be used to pull a membrane tether by applying a harmonic force to a vertex (see Supplementary Movie 1 and Supplementary Fig. 2). Such a system can be used, for example, to study protein sorting[12] or to obtain membrane bending rigidity[46]. Figure 3 illustrates how FreeDTS can be used to pull a membrane tether as well as to study curvature-driven protein sorting that is dependent not only on protein curvature but also on protein–protein interactions.

### Other shapes: vesicles, high-genus membranes and tubes

The type of membranes (surfaces) that can be simulated with FreeDTS is not limited to flat membranes. In principle, any triangulated surface that is closed (even if it is through a period box) can be handled. A prime example is a spherical surface such as vesicles. Supplementary Fig. 3 shows the transition of a spherical vesicle from prolate-to-oblate and oblate-to-stomatocyte by volume reduction[47]. Figure 4A, B also shows how both types of inclusions could induce tubular membrane invaginations in a vesicle.

An important characteristic of membranes is the topology of their surfaces. While membranes are flexible and easy to bend, their surface topology tends to remain constant, as topological changes require membrane fission and fusion processes that are restricted by a high-energy barrier. Topology is characterized by the topological genus $g$, which counts the number of handles attached to a sphere. For instance, $g = 0$ for a sphere and $g = 1$ for a coffee mug. Although organelle membranes, such as the ones found in mitochondria and Golgi apparatus, exhibit high-genus topology, there has been only a limited number of studies conducted on membranes with non-spherical topologies, which have considered only simple, low-genus membranes[48–50]. FreeDTS allows for the exploration of these surfaces. Just as an example, Fig. 4C, D shows how a closed surface with topological genus 20 transformed by some inclusions into a stomatocyte structure. This process might be relevant for nuclear membrane formation and assembly of nuclear pore complex driven by membrane curvature[51].

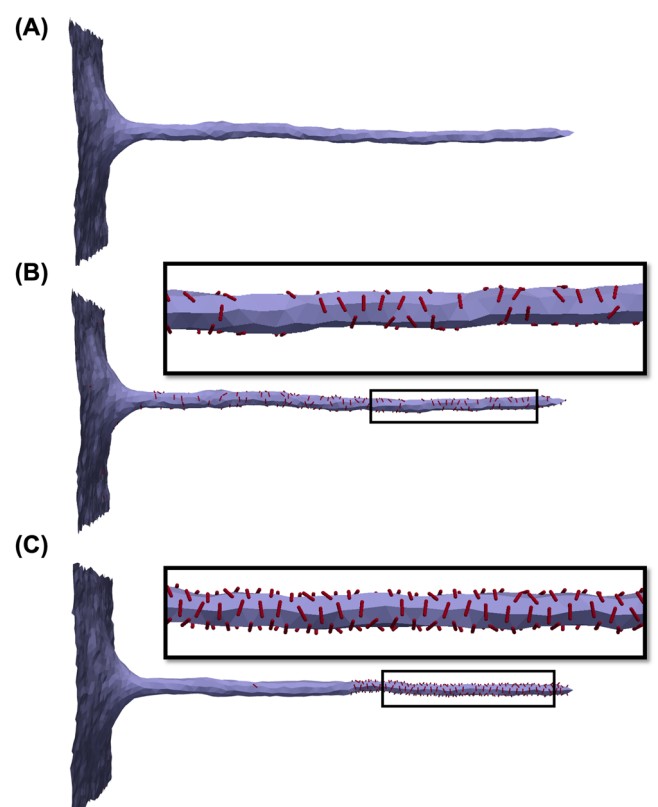

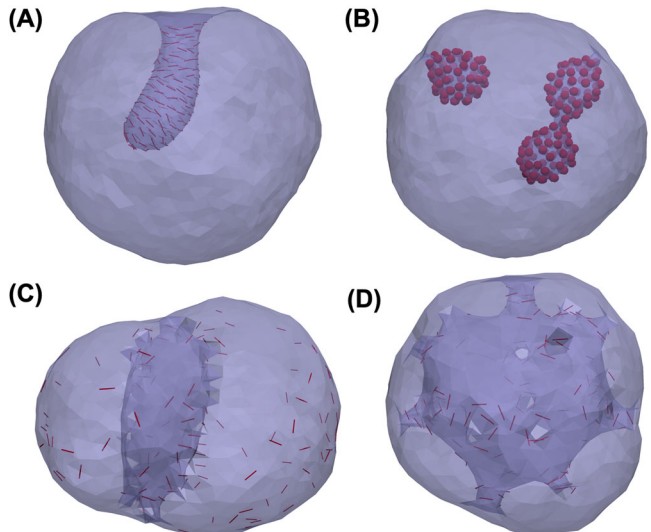

**Fig. 4 | Vesicle and high-genus membranes remodeling by proteins. A** Tubular membrane invagination induced in a vesicle by 20% coverage of protein type 2 with $k_1 = 10[k_BT], k_2 = 5, C_{||0} = 1[l_{dts}^{-1}]$, $C_{\perp 0} = 0, e_{inc,inc} = -k_BT(2 + \cos[2(\Theta)])$. **B** a vesicle containing 20% proteins type one with $\Delta\kappa = \Delta\kappa_G = 0$, $c_0 = 0.6[d_{dts}^{-1}], e_{inc,inc} = -k_BT$. **C, D** a high-genus membrane (g = 20) with $k_1 = 15[k_BT], k_2 = 0, C_{||0} = 1.5[l_{dts}^{-1}]$, $e_{inc,inc} = 0$ (**C**) starting configuration (**D**) final configuration, a minimal model for nuclear membranes where nuclear pore complex assembles and transforms the handles into pores.

**Fig. 3 | Tether pulling from membranes with periodic boundary conditions and constant frame tension. A** A tube is pulled from a framed membrane coupled with a constant tension of $\tau = 2[k_BT l_{dts}^{-2}]$; **B** protein sorting due to curvature. Protein coverage 10%, type 2, with $k_1 = 10[k_BT], k_2 = 0, C_{||0} = 1[l_{dts}^{-1}]$, $C_{\perp 0} = 0, e_{inc,inc} = 0$; **C** protein sorting due to curvature influenced by protein–protein interactions. Protein coverage 10%, type 2, with $k_1 = 10[k_BT], k_2 = 0, C_{||0} = 1[l_{dts}^{-1}]$, $C_{\perp 0} = 0, e_{inc,inc} = -k_BT(1.5 + 0.5 \cos[2(\Theta)])$. The red lines show the proteins and their orientation on the plane of the membrane.

Another type of membrane structure is tube, a surface periodic in one direction (Fig. 5A). Tubes can be used to study processes on the segments of, for example, smooth endoplasmic reticulum. Figure 5 shows how FreeDTS can be used to explore tube deformations by different means.

### Confined membranes

In FreeDTS, membranes can be contained within spaces of various shapes and sizes. In the current version, a membrane can be confined between two sandwiching walls, an ellipsoidal shape, an ellipsoidal shell, or a block (Fig. 1B and Supplementary Fig. 4).

### Protein sorting on realistic membranes

Advances in experimental techniques such as cryo-electron tomography now allows for resolving membrane shape of a full organelle or even a cell[2,52]. When the density and mesoscale parameters of each biomolecule are provided, FreeDTS can use these structures as input files to determine biomolecular organization. Figure 6 shows a DTS simulation of a real inner mitochondrial membrane with two different kinds of proteins (each with different model parameters). It is important to note that these proteins are not representative of realistic mitochondrial proteins and have only been used as a proof of concept. Proteins are sorted in different regions of the membrane based on the type of protein and their interactions with one another. In this simulation, only the proteins have the possibility of organizing and the triangulated mesh does not change (shape change).

## Discussion

Mesoscopic simulations are a necessary tool to explore many important features of membrane-involved cellular processes[20]. In particular DTS approaches have shown to be very robust in capturing spatial and lateral organization of biomembranes. FreeDTS as shown in the current manuscript and our previous works[12,26,31,38] offer a great technology for exploring these processes. Also, as shown in Fig. 6, FreeDTS can be used to obtain the organization of biomolecules on experimentally resolved membrane shapes. In principle, protein density and protein–protein interactions of main membrane-shaping proteins can be obtained from experiments such as cross-linking mass spectrometry[53] or from higher-resolution simulations[54]. Other mesoscopic model parameters (such as induced local curvature) can be obtained from molecular simulations[41] or from a top-down approach through fitting to experimental data in controlled biophysical experiments[12,55]. Once these parameters are available, FreeDTS can be used to obtain the organization of these biomolecules. Furthermore, the equilibrated output of the FreeDTS can be processed directly by TS2CG to create the system structure with a molecular resolution[29,30] for full cell or organelle simulations[11,56].

In a series of pioneering works, Voth and coworkers introduced a mesoscopic membrane model (called EM2, later upgraded to MesM-P)[23,24,57] that can be performed using LAMMPS open-source molecular dynamics package[58,59]. This model has been successfully used to describe complex membrane morphologies induced by BAR-domain proteins. While the model described in this manuscript shares certain similarities with MesM-P, there exist fundamental distinctions. Firstly, in FreeDTS, a membrane is explicitly represented as a surface, and the evolution of this surface is governed by the simultaneous adherence to self-avoidance principles and the preservation of the manifold configuration of the surface. In contrast, in MesM-P, the starting point is a coarse-grained model in fluid dynamics, where the explicit solvent and membrane components are represented by quasi-particles and the membrane's bending energy is included through a particle position with respect to its nearest neighbor membrane particles. Effects of curvature and topology changes e.g., membrane fragmentation, are

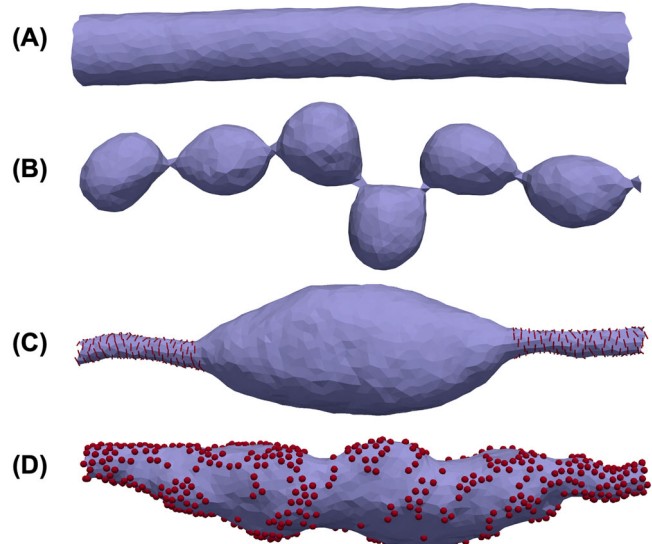

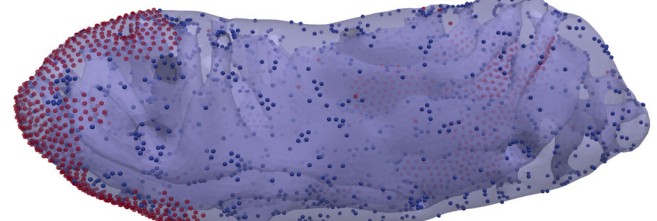

**Fig. 6 | Experimentally resolved real biological membranes can be loaded in FreeDTS to obtain protein organizations.** A real inner mitochondrial membrane containing two different kinds of proteins simulated by FreeDTS. Blue and red spheres are showing the proteins (each color to represent a specific kind of protein).

**Fig. 5 | Configuration of a tube with periodic boundary conditions along its axis. A** A tube made of a simple membrane. **B** A tube with spontaneous curvature ($\bar{C} = 0.4[1/l_{dts}]$). **C** A tube with 20% protein coverage. The proteins are type 2 with $k_1 = 10[k_B T], k_2 = 0, C_{\parallel 0} = 1.5[d_{dts}^{-1}], C_{\perp 0} = 0$ and $e_{inc,inc} = -k_B T(1 + \cos[2(\Theta)])$. **D** A tube decorated with 40% proteins of type 1. $\Delta\kappa = 0, c_0 = 0.4[d_{dts}^{-1}], \Delta\kappa_G = 0$ and $e_{inc,inc} = -1$. The red lines and spheres are the proteins.

described through anisotropic quasi-particle interactions, while they are handled with geometrical quantifiers in FreeDTS. Note, the current version of FreeDTS does not allow for surface topology change, however, it can be achieved by the addition of certain discontinuous Monte Carlo moves (see below)[60,61]. The explicit representation of the membrane surface in FreeDTS is important, in particular, in the modeling of highly curved and folded membranes, e.g., subcellular membranes with tubular or high-genus structures (see Fig. 4D). It also allows for correct measurement of quantities such as system volume, surface area, local curvature, and surface topology and offers several advantages, such as the utilization of numerous algorithms and geometrical descriptions originally developed for image processing techniques providing a wealth of algorithms to adopt for mesoscopic modeling of membranes. Also, FreeDTS allows for a more accurate representation of membrane proteins coupling to membrane curvature making it especially suitable for elongated proteins. In addition, it enables the modeling of proteins that induce changes in the membrane's Gaussian modulus, a factor that has been demonstrated to be critical for the emergence of membrane-mediated interactions. Last but not least, FreeDTS allows for parallel transport which is very important for proper modeling of anisotropic in-plane interactions between membrane proteins on curved surfaces.

To obtain biologically relevant information, models such as DTS may appear to be highly dependent on calibrating their parameters to start with. It should be noted, however, that this is not entirely accurate. Even without any knowledge of membrane-shaping protein structure, DTS simulation can provide some knowledge about their structure by tuning the model parameters against macroscopic biophysical experiments[26,55].

FreeDTS can be expanded (and is in our agenda) in many different ways to become a versatile tool to explore membrane remolding processes. For example, during a simulation in FreeDTS, the number of vertices, triangles, and edges of the triangulated mesh remains constant. In addition, the membrane surface is considered closed, i.e., no possibility of an open edge, such as a hole, exists. Due to these limitations, FreeDTS does not allow for several membrane-related cellular processes, including membrane fission, fusion and membranes with

large holes[62,63]. Fission and fusion can be captured by developing a dynamic topology algorithm such as refs. 60,61,64. It is, however, more challenging for a membrane with open edges, especially since the energy of the edge also depends on the geodesic curvature of the edge[65]. This indeed requires theoretical developments on how to evaluate these geometrical variables, e.g., edge geodesic curvature on triangulated surfaces.

The Helfrich Hamiltonian (the membrane bending energy used by FreeDTS) is a function of principal curvatures up to the second order of principal curvatures assuming the energy should be invariant under in-plane rotation. It has been demonstrated that this bending energy is very excellent for describing large-scale membrane shapes[1]. Nevertheless, at the mesoscale, the effects of higher order may become significant as cellular membranes often exhibit curvature radii that are comparable to the thickness of the membrane[36,66]. However, the challenge of resolving this problem is more theoretical in nature, i.e., finding a suitable bending energy function. Once obtained, its implementation within FreeDTS is rather straightforward without a significant increase in computational cost.

FreeDTS is currently designed to explore the equilibrium shape of complex membranes that is the answer to a wide range of key membrane-involved biological processes. Nevertheless, there are important biological processes that require a detailed description of membrane dynamics, consequently a correct membrane and solvent hydrodynamics, e.g., pearling instability[67], which is beyond the capabilities of the current FreeDTS version. Previously there have been some attempted to include the hydrodynamics effects, using implicit and explicit solvent particle, in both coarse-grained and mesoscopic simulations[68-71]. Capturing realistic dynamics, in particular the effects of long-range hydrodynamics, is a challenging task and demands expensive computations. Nevertheless, the shape operator framework used in FreeDTS make it possible to evaluate in-plane vector and tensor fields, thereby providing a new strategy for coupling surface mechanics with hydrodynamics which in principle could have lower computational cost. Therefore, we expect that in the future, FreeDTS will become capable of handling sufficiently accurate hydrodynamics and dynamics without significant computational costs.

FreeDTS currently runs most movies sequentially and only uses one CPU core (with the exception of parallel tempering, where each replica is run on a separate core). The moves can in fact be parallelized to some extent[28] in order to explore membranes with more vertices or obtain results more quickly for smaller systems. However, this may not lead to better membrane exploration. It is, for example, not amendable to properly capture configurations of membranes larger than a few folds just by allocating more resources, since the required steps to properly sample the configurations increase rapidly (nonlinearly) with system size. In addition, membranes exhibit shapes with very different configurational structures, separated by many large energy barriers, but energetically close (degenerate). Therefore, often it is more practical to perform many replicas to capture these possible

configurations, or use enhanced sampling methods e.g., parallel tempering and Hamiltonian replica exchange methods that are much easier to parallelize efficiently for DTS-like methods.

In biomembranes, different phases with specific molecular compositions coexist. These phases can recruit membrane proteins which have been postulated to be essential for many biological processes such as signaling and endocytosis[72,73]. Currently, FreeDTS is capable of capturing such phenomena, but the total surface area of each phase remains constant. In a biological system, however, these phases are dynamic and can appear, disappear, shrink, or expand depending on the physiological conditions. In the future developments, we will address this limitation.

Finally, the current software can be enhanced by equipping it with models of, for example, cell walls, cortical actin cytoskeleton and active processes (breaking detailed balance) or by coupling the membrane mechanics to kinetic models of the cellular processes to provide a mechano-kinetic model of the cell membranes[74,75].

To summarize, we have presented FreeDTS software to perform computational research on biomembranes at the mesoscale. The model parameters of proteins have physical meaning and in principle can be calibrated using finer scale simulation techniques e.g., all-atom and coarse grain molecular dynamics, or through a top-down approach through experimental data. Several algorithms are included in the software that allows for simulations of framed membranes with constant tension, vesicles with various fixed volume or constant pressure difference, confined membranes into the fixed region of the space, constant fixed global curvature, and application for external forces on regions of the membrane. In addition, the software allows one to turn off the shape evolution of the membrane and only explore inclusions organization. This allows us to take realistic membrane shapes obtained from cryo-electron tomography and obtain heterogeneous organization of biomolecules which can be backmapped to finer simulations models. In addition to its use for exploring many biomembrane application, this software brings us a step closer to simulating realistic biomembranes with molecular resolution.

## Methods
### Membrane model
FreeDTS represents a membrane as a triangulated surface containing $N_v$ vertices, $N_e$ edges and $N_T$ triangles (Supplementary Fig. 5). During the system evaluation, the vertices position gets updated and the mutual link between two adjacent triangulates can be flipped (see below and ref. 54). These two moves allow us to sample through all possible triangulations for a given $N_v$, $N_e$, and $N_T$ and therefore such representation is often referred to as dynamically triangulated surfaces (DTS). DTS is a very popular and successful model to describe shape configurations of interfaces, surfaces, and lipid membranes[76,77]. For the purpose of a mesoscopic model, each vertex represents a membrane patch containing hundreds of lipids[54]. Discrete geometric operations are used to determine the geometric properties of the surface at each vertex. Several methods are available, each with its own advantages and disadvantages[78–80]. In the current version, we are using a method based on Shape Operator described in ref. 81 where the verification is given for well-defined geometries. The reason for this choice is that with this scheme we can obtain, on each vertex $v$, associated principal curvatures ($c_{1,v}, c_{2,v}$), and principal directions ($\hat{T}_{1,v}, \hat{T}_{2,v}$), in addition to an associated area ($A_v$) and surface normal ($\hat{N}_v$) (see Supplementary Note 2 and Supplementary Fig. 5). Moreover, this scheme allows for parallel transport of in-plane vector fields. These quantities are particularly important when modeling anisotropic proteins and protein–protein interactions (see below). These features are well verified in ref. 82 where the simulation results agrees well with theory in analytically tractable limits.

Due to the fluid nature of membranes (the type of membrane that we are considering which encompass most of biological membranes),

bending energy will be a function of mean and Gaussian curvatures. In the current version, we will use a discretized version of the Helfrich Hamiltonian that is a function of surface curvature up to the second order (Eq. (2)),

$$E_b = \sum_{v=1}^{N_v} \left\{ \frac{\kappa}{2} \left( 2H_v - \bar{C} \right)^2 - \kappa_G K_v \right\} A_v \tag{2}$$

where the sum is over all vertices, $2H_v = c_{1,v} + c_{2,v}$, $K_v = c_{1,v}c_{2,v}$ and $\kappa, \kappa_G$ and $\bar{C}$ are bending rigidity, Gaussian modulus, and spontaneous membrane curvature, respectively (the model parameters of the membrane). The second term is written with a minus sign so that $\kappa_G$ is positive. However, often in the literature a positive sign is used for the second term and therefore the reported value of $\kappa_G$ is negative. Equation (2) implicitly indicates that a vertex has the character of a surface element rather than a particle. As a note, different bending energy functions can be easily adopted in FreeDTS without any significant performance reductions.

To ensure self-avoidance, there is a hard-core potential between the vertices such that the minimum distance between any two vertices must be equal to $l_{dts}$ (the basic length unit in FreeDTS). In addition, self-avoidance requires that the edge length vary within a specific range ($l_{min} \le l_e \le l_{max}$). It has been tested that $l_{min} = l_{dts}$ and $l_{max} = \sqrt{3}l_{dts}$ with a mild constraint on the dihedral angle between two neighboring triangles is enough to ensure self-avoidance of the surface[81]. As a note, $l_{min}$, $l_{max}$ and the minimum dihedral angle can be set by the user in FreeDTS.

Due to the flexibility of the edge size, the area of each triangle (in principle) can vary between $\sqrt{3}l_{min}^2/4 \le A_T \le \sqrt{3}l_{max}^2/4$, but on average the total area of the surface remains constant (see Supplementary Fig. 6 and Supplementary Table 1). However, to control the surface area, an area constraint algorithm can be loaded which couple the system energy to an additional term as Eq. (3),

$$E_A = N_T \frac{K_A}{2} \left( \frac{A}{A_0} - 1 \right)^2 \tag{3}$$

where $A_0$ is the targeted area and $K_A$ is compressibility modules.

For membrane patches in a periodic box, i.e., with periodic boundary condition (PBC), see Supplementary Fig. 7, the possibility of the box change (dynamic box) is crucial to allow membrane shape remodeling. FreeDTS is equipped with a tension-controlling algorithm that allows for performing simulation at constant (or zero) tension (($N_v, \tau, T$) ensemble). This algorithm couples the system energy to $E_\tau = -\tau A_p$ where $\tau$ is frame tension and $A_p$ is the projected area. We have described the details of this algorithm in ref. 38 and the Supplementary Note 3. Supplementary Fig. 1 shows that this tension ($\tau$) is equal to the tension derived from the undulation spectrum, consistent with previous works[38,83,84].

Due to the osmolarity effect, closed surfaces, such as vesicles, will be bound by the volume of the media they encompass. It is possible to apply an energy potential as Eq. (4), which supports both first- and second-order couplings of the volume in FreeDTS,

$$E_v = -\Delta P V + \frac{K}{2} \left( \frac{V}{V_0} - v_t \right)^2 \tag{4}$$

where $\Delta P$ is the pressure difference between in and outside, $V$ is the vesicle volume, $K$ is volume compressibility modulus (coupling constant), $V_0 = \frac{1}{6\pi^{1/2}} A^{3/2}$ i.e., the volume of a perfect sphere with the area of $A$ (area of the triangulated surface), and $v_t$ is the targeted reduced volume ($0 < v_t \le 1$).

FreeDTS also provide a more sophisticated volume coupling through the Jacobus van 't Hoff equation to better model osmotic

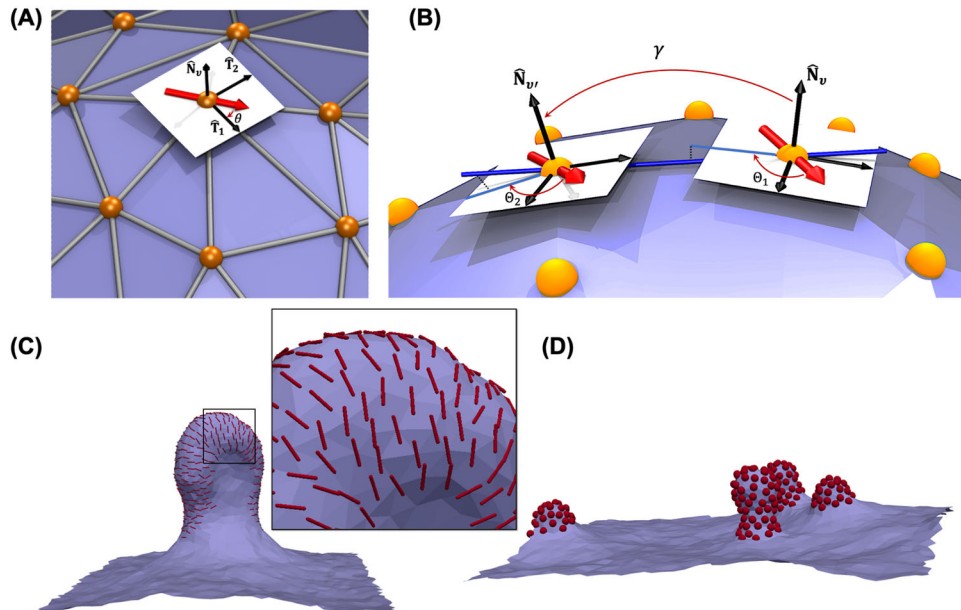

**Fig. 7 | A protein is defined as an in-plane inclusion. A** An inclusion has a varying orientation in the plane of the vertex. $\theta$ is the angle between the inclusion orientation and the first principal direction ($\hat{\mathbf{T}}_1$). **B** Inclusions interact with one another if they are located on two neighboring vertices through a potential in Eq. (10). $\Theta_1$ and $\Theta_2$ are the angle between each inclusion with the projection of a line (blue line) connecting the two vertices. For parallel transport, each inclusion orientation must remain constant with respect to these projections. $\gamma$ is the angle between the two normal vectors ($\hat{\mathbf{N}}_v$, $\hat{\mathbf{N}}_{v'}$). But defined as positive if the tip-to-tip distance is larger than the edge length and negative otherwise. **C, D** Protein–Protein interaction influence the membrane shape and organization of proteins (**C**) A membrane with periodic boundary condition and 20% protein coverage of type 2 with $k_1 = 10\,[k_B T]$; $k_2 = 0$; $C_{\|0} = 1[l_{dts}^{-1}]$; $C_{\perp 0} = 0$ and $A_{i,j} = B_{i,j} = 1[k_b T]$; $C_{i,j} = 0$; $\Theta_0 = 0$. The red lines show the location of the proteins and their orientation on the plane of the membrane (**D**) A membrane with periodic boundary condition and 10% protein coverage (of type 1) without any local curvature ($\Delta\kappa = c_0 = \Delta\kappa_G = 0$) induces membrane budding solely due to protein–protein interaction $A_{i,j} = B_{i,j} = 0$; $C_{i,j} = 20\,[k_b T]$ and $\gamma_0 = \pi/4$. The red spheres are the proteins on the membrane surface.

pressure. In this way, the energy of the system is coupled to Eq. (5),

$$\Delta E_{osmos}(V) = -RT\left[\bar{c}_{in}V_{ini}\ln\frac{V}{V_{ini}} - \bar{c}_{out}(V - V_{ini})\right] \tag{5}$$

where $V_{ini}$ is the initial volume and $\bar{c}_{in}(\bar{c}_{out})$ is effective concertation of the inside (outside) solute (more details can be found in Supplementary Note 4).

While Eq. (2) allows for controlling the spontaneous membrane curvature, spontaneous membrane curvature could be global, originating from, for example, the number of lipid molecules in each monolayer.

FreeDTS allows for controlling global membrane curvature energy to a potential as

$$E_s = \frac{k_r}{2A}\left(M - m_0 A\right)^2, \tag{6}$$

where $M = \sum_{v=1}^{N_v} 2H_v A_v$, $m_0$ is the average membrane global curvature, and $k_r$ is the coupling constant. Since the area difference between inner and outer membrane ($\Delta A$) is proportional to $M$ ($\Delta A = hM$), Eq. (6) can also be used to control area difference.

### Protein model

Proteins are modelled as in-plane inclusions with an in-plane orientation ($\hat{\mathbf{D}}$, also see Fig. 7A). They interact with the membrane and locally modify the membrane physical and mechanical properties e.g., spontaneous curvature and rigidity. There is at most one inclusion per vertex, which naturally handles the in-plane excluded volume effect between inclusions. Inclusions also interact with one another (see below). In FreeDTS two types of membrane-inclusion interactions exist. First are for proteins with symmetric (or almost symmetric) interactions (Supplementary Fig. 8A), such as the one seen for Shiga

and cholera toxins[40,41]. The interaction energy between a vertex and an inclusion of this type is given by Eq. (7).

$$e_{mem-inc} = \left\{2\Delta\kappa H_v^2 - 2(\Delta\kappa + \kappa)c_0 H_v - \Delta\kappa_G K_v\right\}A_v \tag{7}$$

Where $H_v$, $K_v$ and $A_v$ are mean curvature, gaussian curvature and area associated with the vertex $v$ and $\Delta\kappa$, $c_0$ and $\Delta\kappa_G$ are the inclusion model parameters. These parameters has physical meaning, $\Delta\kappa$ and $\Delta\kappa_G$ can be seen as an increase in the membrane bending rigidity and gaussian modulus and $c_0$ as a local curvature (for a discussion on this, see ref. 8).

Second protein type are the one which break in-plane symmetry of the membrane (Supplementary Fig. 8B), such as dynamin and BAR protein family. Membrane-protein interaction between this type of inclusions is given by Eq. (8).

$$e_{mem-inc} = \left\{\frac{k_1}{2}\left(C_{\|} - C_{\|0}\right)^2 + \frac{k_2}{2}\left(C_\perp - C_{\perp 0}\right)^2\right\}A_v \tag{8}$$

$C_{\|}$ and $C_\perp$ are membrane curvature in a direction parallel and perpendicular to the inclusion orientation. Since the used discretized geometric operations allows us to obtain $\hat{\mathbf{T}}_1, \hat{\mathbf{T}}_2, c_1$ and $c_2$, we can obtain $C_{\|}$ and $C_\perp$ using Eulers curvature formula as

$$\begin{cases} C_{\|} = c_1\cos^2\theta + c_2\sin^2\theta \\ C_\perp = c_1\sin^2\theta + c_2\cos^2\theta \end{cases} \tag{9}$$

where $\theta$ is the angle between the orientation of the inclusion and the first principal direction ($\cos\theta = \hat{\mathbf{D}}.\hat{\mathbf{T}}_1$, also see Fig. 7A). $k_1, k_2, C_{\|0}, C_{\perp 0}$ are protein model parameters. $k_1$ and $k_2$ are the directional bending rigidities imposed by the inclusion on the membrane. $C_{\|0}$ ($C_{\perp 0}$) is inclusion preferred membrane curvature in the direction of (perpendicular to) its in-plane orientation (for a discussion on this, see refs. 8,54). Note

Eqs. (7) and (8) indicate that the effective interaction area of an inclusion with the membrane is $A_v$. Therefore, knowing the size of the protein will convert $l_{dts}$ to a physical unit such as nm[12,40] (also see Supplementary Note 1). The model, however, requires some modification of the excluded volume contributions when dealing with small proteins (smaller than membrane thickness, <4 nm). In this case, a vertex could in principle own more than one inclusion, and therefore the excluded volume should be explicitly introduced into the system energy and $l_{dts}$ will be defined by maximum number of inclusions occupying a vertex.

Protein–protein interactions will be a function of the angle between the two inclusions along the geodesic ($\Theta = \Theta_2 - \Theta_1$ and see Fig. 7B) and the normal angle between the vertices that the proteins occupy ($\cos\gamma = \pm\hat{\mathbf{N}}_{v1}.\hat{\mathbf{N}}_{v2}$, positive if tip-to-tip distance is larger than the edge length and negative otherwise). The interaction between two neighboring inclusions $i$ and $j$ can be expressed as a Fourier expansion at the lowest order as

$$e_{inc,inc}(\Theta,\gamma) = -A_{i,j} - B_{i,j}\cos[n(\Theta - \Theta_0)] - C_{i,j}\cos[\gamma - \gamma_0] \quad (10)$$

The first term ($A_{i,j}$) models the isotropic part of the interaction, while the second term models anisotropic interactions (Fig. 7C). The third term allows for additional membrane curvature imprint by two proteins (only if they aggregate) related to the value of $\gamma_0$ (see Fig. 7B). $n$ is the least common multiple of the degree of the $i,j$ proteins symmetry in the plane of the membrane. $A_{i,j}$, $B_{i,j}$ and $C_{i,j}$ are strength coefficients of each energy term. $\Theta_0$ is phase shift. The third term in Eq. (10) models proteins that induce membrane curvature through dimerization or oligomerization (see Fig. 7D)[85,86]. This interaction energy (Eq. (10)) can also be used to model lipid domain formations in multicomponent membranes. FreeDTS can easily accommodate a more complicated interaction function[54], but for the moment, Eq. (10) suffices within the resolution of the mesoscopic model.

### System evolution
System evolution and the equilibrium properties of the membranes are evaluated by Monte Carlo sampling of Boltzmann's probability distribution. Every Monte Carlo step consists of, $N_T$ Alexander moves, a trial to flip the mutual link between two neighboring triangles, $N_v$ vertex positional updates and $N_i$ inclusion moves (where $N_i$ is the number of the inclusions in the system)[54]. If a system is coupled to a tension-controlling scheme, then, with the a given probability, there will be also one trial move to change the box size (see Supplementary Note 3)[38]. There is also a possibility to activate parallel tempering algorithm. This will run multiple replicas at different temperature using OpenMP parallelization to better sampling of the system (for more detail, see Supplementary Note 5).

### About the source code
FreeDTS is implemented in C++. It is self-contained and does not depend on any additional library. It is very well-linked to TS2CG through input and output meshes and can be expanded to be linked with TS2CG on the fly. Also, it is easy to compile and run (for more detail, see the tutorial section in the manual or the tutorial section on GitHub). The energy unit is $k_BT$ and the unit length is $l_{dts}$ which is the minimum distance between any pair of vertices. For information about the code performance, see Supplementary Note 6.

The source code can be found at https://github.com/weria-pezeshkian/FreeDTS.git.

### Reporting summary
Further information on research design is available in the Nature Portfolio Reporting Summary linked to this article.

## Data availability
Any other files related to this manuscript are available from the corresponding authors upon reasonable request. Source data are provided with this paper.

## Code availability
The source code data generated in this study have been deposited in the Github and Zenodo databases under accession codes https://github.com/weria-pezeshkian/FreeDTS.git and https://doi.org/10.5281/zenodo.10397542[87], respectively.

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

## Acknowledgements

W.P. acknowledges funding from the Novo Nordisk Foundation (grant No. NNF18SA0035142 and NNF22OC0079182) and Independent Research Fund Denmark (grant No. 10.46540/2064-00032B). We thank Siewert-Jan Marrink for instructive discussions.

## Author contributions

W.P. programmed the code and obtained the results. W.P. and J.H.I. wrote the manuscript. All authors discussed the results and commented on the manuscript at all stages.

## Competing interests

The authors declare no competing interests.
