## [Peer Review File · Nature Communications]

REVIEWER COMMENTS

Reviewer #1 (Remarks to the Author):

The manuscript “Mesoscale simulation of biomembranes with FreeDTS” reports a new computational tool for efficiently simulating the biological membrane using dynamically triangulated surfaces (DTS). The DTS model is an important tool for simulating the dynamics of biological membranes on lengthscales inaccessible to molecular dynamics calculations where lipids must be incorporated explicitly. The core advance of this manuscript is the presentation an open-source package – FreeDTS – which implements a DTS model with a variety of features. The development of an open-source package will provide a valuable tool for a wide range of communities interested in the dynamics of biological membranes. With the continued development of membrane-to-cellular scale structural measurements using tools such as cryo-electron tomography, FreeDTS is likely to be a useful and important addition to the field.

The manuscript could be substantially improved by a more careful introduction of the relevant equations. Here are some of the problems that made understanding the current model more challenging in my first reading:

1. variable definitions are sometime unclear or placed far from the first use
 - a. A_v is only defined in the SI but appears in main text eq. 1 (the same problem appears for c_1 and c_2)
 - b. Eq. 1. Uses H_v but the following phrase only defines H
 - c. Eq. 3 appears to use E_{ν} to represent the energy associated with the targeted reduced volume (ν), but the symbol used is either the same or almost indistinguishable from the one used for indexing vertices.
 - d. Eq. 6 uses K but the following phrase defines a K_v that does not seem to appear in the equation
2. indexing within the equations is missing from some equations
 - a. Eq. 1 uses a summation but the symbol being summed over is missing (presumably ν)
 - b. Eq. 1 appears to have implicit ν dependence in several variables (e.g. c_1 and c_2).
 - c. Eq. s2 includes a sum over the faces in a ring around a vertex (\sum_R), but then how terms like n_e (the normal associated with an edge) should be handled is not clearly indicated since each face around the vertex will have two edges associated with it.

I would suggest the authors consider taking a more verbose approach to introducing their equation with a focus on making it readable to people who have not previously used DTS models.

Given that the manuscript primarily deals with the presentation of a new software package, it would also be useful to give an overview of how the code base is structured and any key algorithmic developments/choices. This seems particularly pertinent since the current github page and user guide do not introduce the code base and are focused instead on how to use the package. While this introduction of the underlying code does not need to be lengthy, a clear explanation of how the code is structured will increase the value of the open-source software for the larger community by reducing the time required for other groups to modify the code when needed.

Finally, the manuscript presents a number of results in Fig. 2-6, but provides relatively little context for understanding the computational challenge of the tasks being demonstrated. The questions: “how efficient is this implementation?” and “how does this implementation compare to previous DTS codes?” are not currently answered and this makes it difficult for a potential user to assess the relative merits of this code as a computational tool. Some additional questions to consider are: What defines a challenging versus a simple calculation? What are the current limits on feasible calculations using the code base and a modern high performance computing cluster?

Reviewer #2 (Remarks to the Author):

The authors present a freely distributed software FreeDTS, which can be used to simulate fluid membranes, and some types of coarse-grained embedded proteins and their interactions. The manuscript provides a number of examples to demonstrate the capabilities of the code, though most (if not all) of those have already been published elsewhere. The main advantage of this software is that it is freely available, however, its performance is much less clear, as it currently runs only on a single CPU. Furthermore, it is based on Monte Carlo, suggesting that a study of membrane dynamics is very much limited.

I assume this code can be used for studying a range of systems involving biomembranes, but it is impossible to comment on whether it will be used by a significant number of researchers. Furthermore, I have a few concerns which are described below.

1) As I mentioned above, the performance of the code is not very clear. I think the authors should provide examples of simulated system sizes, running lengths, and times, so that it would be possible to estimate performance limitations. What system sizes would become difficult to do?

2) Another aspect is the possible parallelization, which the authors mention, but it may not always be straightforward, as this often requires the locality of interactions. Have you thought about it already? Also, possible use of accelerators?

3) For membranes, bending rigidity is important, requiring a proper discretization. It was not clear what you employ, but there exist studies of that, e.g. Guckenberger & Gekle, J Phys: Condens Matter 29:203001 (2017). How well is your discretization verified?

4) There is a software called OpenRBC, which claims to have molecular resolution. I am not fully sure what it is capable of, but is that something similar?

5) What advantages/disadvantages Monte Carlo has over the integration like molecular dynamics? Why did you select MC?

6) You mention that cytoskeleton can be added to the code. My feeling is that this would require parallel implementation or the use of accelerators, as the system size might become prohibitively large.

Reviewer #3 (Remarks to the Author):

The authors study biomembranes at mesoscopic scales using a certain software called FreeDTS. The manuscript has a section on "The model" followed by a "Results" section. My main points of criticism are ordered in the same way:

First about "The Model" section:

1) The abbreviation FreeDTS appears already in the title. In the abstract, it is stated that FreeDTS is some software. However, the authors do not explain the meaning of this abbreviation. Neither do they explain who developed this software.

2) The method used here describes the lipid membranes as triangulated surfaces which are studied by Monte Carlo simulations. The vertices of the triangulation are decorated by "in-plane inclusions" representing membrane proteins. One confusing aspect about the simulation approach

described here is the basic length scale used in this approach. On page 2, beginning of last paragraph, the edge length l_e of the triangles is introduced, which seems to provide the basic length scale of the approach. However, the length scale l_e is not mentioned in the figure captions of Figs. 2 - 6 where we find instead the length scale d_{dts} .

3) It requires a systematic search through the whole main text to find out how d_{dts} has been defined. On page 6, first paragraph, one finds the statement that "the unit length is d_{dts} , which is the minimum distance between any pair of vertices". However, it is completely unclear why the authors need the new notation d_{dts} because they introduced the minimum distance l_{\min} of the edge length already on page 2, where they also set $l_{\min} = l_{dts}$. As a result, the manuscript uses three different notations - l_{\min} , l_{dts} , and d_{dts} - for the same basic length scale, which is very confusing and must be avoided.

4) In addition to the notational confusion just described, the authors do not seem to make any attempt to estimate this length scale in terms of physical units, that is, in terms of nanometers. However, without such an estimate, it is hardly possible to assess the membrane conformations displayed in the figures of the "Results" section.

5) In equation (1), the Gaussian curvature modulus is written with a minus sign which differs from the standard definition of this modulus.

6) The lateral size of the membrane proteins is not taken into account because these proteins are spatially confined to the vertices of the triangles, that is, they are described as point-like particles with some internal degrees of freedom corresponding to their orientation. Apparently, the authors assume here that the lateral size of the membrane proteins is small compared to the (average) edge length of the triangles but this assumption should be spelled out explicitly. Furthermore, this type of modeling becomes problematic when the lateral size of the proteins is large compared to the lateral size of the lipids, the latter being typically between 0.5 and 0.7 nm.

7) The interaction between two proteins on two neighboring vertices is described by equation (9). This interaction, which is claimed to represent "the simplest interaction between two neighboring inclusions", depends on five model parameters. The numerical values chosen for these parameters are mentioned in some of the figure captions but it remains completely unclear where these numerical values come from.

8) The Monte Carlo simulations described here do not take the aqueous solutions surrounding the membrane into account. Therefore, these simulations ignore the hydrodynamics of these solutions as well as the resulting hydrodynamic interactions between different membrane segments. These

hydrodynamic interactions affect the relaxation of membrane undulations and protein orientations. Hydrodynamic interactions are also crucial during the time-dependent shape transformations from one membrane morphology to another. Therefore, the computational approach described here does not lead to a realistic dynamics of membranes and proteins.

9) The Monte Carlo approach has been previously extended to include hydrodynamics. One such extension is by Noguchi and Gompper in Phys. Rev. Lett. 93, 258102 (2004), which is based on a combination of multiparticle collision dynamics with Monte Carlo sampling.

10) Likewise, coarse-grained molecular dynamics simulations such as Dissipative Particle Dynamics (DPD) have been developed, which conserve momentum locally and provide a reliable description for the hydrodynamics of both membranes and aqueous solutions. Recent insights into the behavior of membranes and vesicles as obtained by DPD have been reviewed in Lipowsky et al, Biomolecules 13, 926 (2023).

11) The authors emphasize the "frame tension" τ which is modeled by an additional energy term. This term is taken to be $-\tau A_p$, see page 3, first paragraph, where A_p represents the projected area. I am rather sceptical that the authors really want to include a minus sign here because positive values of τ would then lead to membrane compression rather than to membrane stretching.

12) In addition to the sign problem of the frame tension term, the magnitude of this tension is ill-defined when we consider the membranes of vesicles which represent the most popular membrane model systems. As a consequence, it is not clear how the authors derive the numerical values of τ as used in the manuscript, see, e.g., caption of Fig. 4.

Second, about the "Results" section:

13) Because of point 4 above, the size of the membrane morphologies displayed in Figs. 3 - 7 is unclear. The authors need to explain which experimentally accessible membranes are proposed to attain these morphologies on mesoscopic scales.

14) Mesoscopic scales can be probed by light microscopy. One very popular and highly useful membrane system is provided by giant unilamellar vesicles (GUVs). However, when observed by light microscopy, GUVs have smoothly curved

membranes which look very different from the kinky shapes displayed in Figs. 3 - 7.

This kinky appearance must be an artefact of the underlying triangulation of the membrane surfaces. Some snapshots appear to be relatively smooth. Did the authors apply some algorithm to smoothen these snapshots?

15) Many of the shapes in Fig. 3 involve protein-rich and protein-poor membrane segments. These shapes represent examples for phase separation into two types of intramembrane domains which then undergo domain-induced budding. The latter process depends on the line tension of the domain boundary which can dominate the budding process as predicted theoretically in Lipowsky, J. Phys. II France 2, 1825 (1992) and observed by several experimental groups. The authors should explain why they chose to ignore this line tension.

A final and general comment:

16) The authors mention several alternative simulation approaches in the introduction of the manuscript but they do not explain why we need yet another simulation approach based on FreeDTS. In addition, they do not compare their results with the results of previous simulations. Such a comparison is necessary, however, in order to assess the advantages and disadvantages of the FreeDTS approach.

We would like to thank the reviewers for their thorough reading of our manuscript, their suggestions and criticisms. We have carefully considered the points raised and modified the manuscript accordingly. We hope that our responses are fulfilling. The referee comments are in *italic* and our responses are in **bold (black-bold our response and blue-bold are the changes in the manuscript that is rewritten in this document). All the modified texts in the main manuscript and the Supplementary information are highlighted in red.**

Additionally, the following publications have been added to the list of references:

- 1) R. Lipowsky, Budding of membranes induced by intramembrane domains, J. Phys. II France 2 (1992) 1825-1840.
- 2) A. Guckenberger, S. Gekle, Theory and algorithms to compute Helfrich bending forces: a review, J Phys Condens Matter 29(20) (2017) 203001.
- 3) Haoran Ni and Garegin A. Papoian J. Phys. Chem. B 2021, 125, 10710–10719; Membrane-MEDYAN: Simulating Deformable Vesicles Containing Complex Cytoskeletal Networks.
- 4) Mohsen Sadeghi and Frank Noé; Large-scale simulation of biomembranes incorporating realistic kinetics into coarse-grained models; Nature Communications 11, 2951 (2020)
- 5) Hiroshi Noguchi and Gerhard Gompper; Fluid Vesicles with Viscous Membranes in Shear Flow PRL 93, 258102 (2004)
- 6) Lipowsky et al, Leaflet Tensions Control the Spatio-Temporal Remodeling of Lipid Bilayers and Nanovesicles; Biomolecules 13, 926 (2023)
- 7) L. Gao, J. Shillcock, R. Lipowsky, Improved dissipative particle dynamics simulations of lipid bilayers, J Chem Phys 126 015101(2007)
- 8) R. Bar-Ziv, T. Tlusty, E. Moses, Critical Dynamics in the Pearling Instability of Membranes, Physical Review Letters 79(6) (1997) 1158-1161.

Reviewer #1 (Remarks to the Author):

The manuscript “Mesoscale simulation of biomembranes with FreeDTS” reports a new computational tool for efficiently simulating the biological membrane using dynamically triangulated surfaces (DTS). The DTS model is an important tool for simulating the dynamics of biological membranes on lengthscales inaccessible to molecular dynamics calculations where lipids must be incorporated explicitly. The core advance of this manuscript is the presentation an open-source package – FreeDTS – which implements a DTS model with a variety of features. The development of an open-source package will provide a valuable tool for a wide range of communities interested in the dynamics of biological membranes. With the continued development of membrane-to-cellular scale structural measurements using tools such as cryo-electron tomography, FreeDTS is likely to be a useful and important addition to the field.

We would like to thank the referee for the positive opinion on our manuscript.

The manuscript could be substantially improved by a more careful introduction of the relevant equations. Here are some of the problems that made understanding the current model more challenging in my first reading:

We would like to thank the reviewer for detailed reading of all equation and methodology. We now have made substantial changes in the presentation of the equation to address all the related comments. Below the related specific change for each comment is mentioned.

1. variable definitions are sometime unclear or placed far from the first use

- a. A_v is only defined in the SI but appears in main text eq. 1 (the same problem appears for c_1 and c_2)
- b. Eq. 1. Uses H_v but the following phrase only defines H

We have changed the below sentence to include the variable definition and also change the presentation of the equation 1 to address this confusion.

The reason for this choice is that with this scheme we can obtain, on each vertex v , associated principal curvatures ($c_{1,v}, c_{2,v}$), and principal directions ($\hat{T}_{1,v}, \hat{T}_{2,v}$), in addition to an associated area (A_v) and surface normal (\hat{N}_v).

- c. Eq. 3 appears to use E_{ν} to represent the energy associated with the targeted reduced volume (ν), but the symbol used is either the same or almost indistinguishable from the one used for indexing vertices.

We now have used v_t for targeted reduced volume to avoid confusion of similarity between v and ν

- d. Eq. 6 uses K but the following phrase defines a K_v that does not seem to appear in the equation

K has been changed to K_v

- 2. indexing within the equations is missing from some equations

- a. Eq. 1 uses a summation but the symbol being summed over is missing (presumably ν)

This has been added now.

- b. Eq. 1 appears to have implicit ν dependence in several variables (e.g. c_1 and c_2).

This has been fixed now.

- c. Eq. s2 includes a sum over the faces in a ring around a vertex (\sum_R), but then how terms like n_e (the normal associated with an edge) should be handled is not clearly indicated since each face around the vertex will have two edges associated with it.

We rewrote this section to make it clearer.

I would suggest the authors consider taking a more verbose approach to introducing their equation with a focus on making it readable to people who have not previously used DTS models.

Given that the manuscript primarily deals with the presentation of a new software package, it would also be useful to give an overview of how the code base is structured and any key algorithmic developments/choices. This seems particularly pertinent since the current github page and user guide do not introduce the code base and are focused instead on how to use the package. While this introduction of the underlying code does not need to be lengthy, a clear explanation of how the code is structured will increase the value of the open-source software for the larger community by reducing the time required for other groups to modify the code when needed.

Following the reviewer recommendation, we created a code map that shows how the main objects of the source code are structured. This file is on the GitHub page as well.

Finally, the manuscript presents a number of results in Fig. 2-6, but provides relatively little context for understanding the computational challenge of the tasks being demonstrated. The questions: "how efficient is this implementation?" and "how does this implementation compare to previous DTS codes?" are not currently answered and this makes it difficult for a potential user to assess the relative

merits of this code as a computational tool. Some additional questions to consider are: What defines a challenging versus a simple calculation? What are the current limits on feasible calculations using the code base and a modern high performance computing cluster?

Following the referee recommendation, we have added “Section 5: performance of the code for the Monte Carlo moves” that contains a description a table and four figures.

Regarding comparing it to other software:

FreeDTS is an efficient implementation. As most of the other implementations are not available, here we make a comparison with the TriMem software data available. Below is a copy of Figure 3 at the bottom left of the original TriMem paper. A system with 642 vertices contains 1280 edges. Based on this figure, on 16 threads, TriMem makes 10×1280 edge flip attempts in 10 milliseconds. According to Figure SI-5.1 and Table SI-5.1, FreeDTS makes the same number of edge-flipping attempts in around 6 milliseconds on a single thread. This clearly shows the efficient implementations of FreeDTS. However, we prefer not to include this comparison in the manuscript and prefer to make a rigorous comparison in collaboration with TriMem developers.

It is worth mentioning that the scheme used in FreeDTS differs from TriMem and the bulk of DTS simulation schemes. FreeDTS scheme makes it possible to explore the conformational properties of membranes with more complex in-plane interactions and inclusions with non-symmetric curvature food-print, and orientational spontaneous curvature. It also makes it possible to operate with in-plane vector and tensor order parameter fields and their coupling the curvature, including features like parallel transport and directional and geodesic curvatures. While these calculations are more expensive, they are essential to create a versatile mesoscopic membrane model. Therefore, it is worth the cost.

Reviewer #2 (Remarks to the Author):

The authors present a freely distributed software FreeDTS, which can be used to simulate fluid membranes, and some types of coarse-grained embedded proteins and their interactions. The

manuscript provides a number of examples to demonstrate the capabilities of the code, though most (if not all) of those have already been published elsewhere. The main advantage of this software is that it is freely available, however, its performance is much less clear, as it currently runs only on a single CPU. Furthermore, it is based on Monte Carlo, suggesting that a study of membrane dynamics is very much limited.

We would like to make it clear that, while many (not all) of the algorithms employed in the code have been published elsewhere (by the authors and others), this software combines these algorithms in one setting allowing us to explore much more complex biomembrane in many different physical conditions. Moreover, while the results presented in the main manuscript are proof-of-concept, they are not published before.

Regarding the performance: please refer to the answer to the last question of reviewer one and new added Section 5 in the SI (related to question 1).

Regarding membrane dynamics: please refer to our response to question 5 and question 8 of reviewer 3.

I assume this code can be used for studying a range of systems involving biomembranes, but it is impossible to comment on whether it will be used by a significant number of researchers.

According to the positive response received from the computational community after publishing the preprint and making the code available, the number of users is likely to be significant. Further, such software is necessary for further progress in the computational exploration of biomembranes (see for example Biophysical Journal 122, 1883 (2023)), and currently no software with this capacity is freely available.

Furthermore, I have a few concerns which are described below.

1) As I mentioned above, the performance of the code is not very clear. I think the authors should provide examples of simulated system sizes, running lengths, and times, so that it would be possible to estimate performance limitations. What system sizes would become difficult to do?

We now added Section 5 in the SI that contains performance analysis of the simulation steps. Also please see our response to the last point raised by reviewer 1.

2) Another aspect is the possible parallelization, which the authors mention, but it may not always be straightforward, as this often requires the locality of interactions. Have you thought about it already? Also, possible use of accelerators?

Parallelization is indeed a challenging task. We have thought about this for future distributions. All the DTS interactions (unlike coulomb interaction for particle-based simulations) are local, and mostly dependent on the state of the neighboring vertices (except for the parallel transport which is dependent on the second neighboring vertices).

Within the MC framework (related to question 5, for MD schemes there are more flexibilities), there are two main challenges, (1) ensuring detailed balance and (2) reproducibility of the simulation trajectory. Within one MC step (unlike MD), changing the sequence of the moves affects the outcome. Therefore, for two neighboring vertices that are being updated by two different threads, the physical speed of the threads (physical state of the threads) affects the outcome (reproducibility) which could (in some cases) also break the detailed balance. A remedy to this is that prior to each move the sequence of the moves must be determined and the threads at the boundary must stay idle until the neighbors are updated (as a note, for box update move such a problem does not exist). Implementation of such an algorithm for shared-memory architectures is rather straightforward (this can be done by only changing the simulation class without the need for any change of the other important classes). However, on distributed memory, several of the FreeDTS data structures must be modified.

We envision that using accelerators would not enhance much since the moves, curvature, and energy calculations require a sequence of dependent calculations.

If the reviewer agrees, we would like to avoid adding any content to the manuscript regarding our response to this question as it is beyond the aim of the current manuscript and involves untested technicalities.

3) For membranes, bending rigidity is important, requiring a proper discretization. It was not clear what you employ, but there exist studies of that, e.g. Guckenberger & Gekle, *J Phys: Condens Matter* 29:203001 (2017). How well is your discretization verified?

The discretization of membrane curvature is based on the Shape operator method introduced in [35], where the verification is given for well-defined geometries. The parallel transport algorithm is probably best verified in [36], where the simulation results agree very well with theory in analytically tractable limits.

To clarify this further we extended the model section of the manuscript that reads as below text (the relevant part for this question) and the paper suggested by the reviewer was cited.

Discrete geometric operations are used to determine the geometric properties of the surface at each vertex. Several methods are available, each with its own advantages and disadvantages [32-34]. In the current version we are using a method based on Shape Operator described in [35] where the verification is given for well-defined geometries. The reason for this choice is that with this scheme we can obtain, on each vertex v , associated principal curvatures $(c_{1,v}, c_{2,v})$, and principal directions $(\hat{T}_{1,v}, \hat{T}_{2,v})$, in addition to an associated area (A_v) and surface normal (\hat{N}_v) (see SI section 1 and Figure SI-8.1). Moreover, this scheme allows for parallel transport of in-plane vector fields. These quantities are particularly important when modeling anisotropic proteins and protein-protein interactions (see below). These features are verified in [36] where the simulation results agrees well with theory in analytically tractable limits.

4) There is a software called OpenRBC, which claims to have molecular resolution. I am not fully sure what it is capable of, but is that something similar?

OpenRBC is a coarse grained molecular dynamics algorithm. The only small resemblance with FreeDTS is that the spectrin cytoskeleton of Red Blood Cell is modelled as a triangular mesh (not dynamic) in OpenRBC with a restricted conformational space.

5) What advantages/disadvantages Monte Carlo has over the integration like molecular dynamics? Why did you select MC?

We assume that the reviewer's question is about employing MD methodology to integrate DTS model since it is clear (also pointed out in the Introduction) that all-atom or even coarse-grained MD are very far from being able to capture membrane shape remodeling processes (apart from some limited cases).

Both MD and MC are rigorous schemes to explore the thermodynamic behavior of biomaterials (and beyond). The debate about MC vs MD is more generic and beyond the current manuscript. However, for the case of the DTS models, hybrid schemes, using MD for:

- a) vertex position updates
- b) inclusion orientation updates

and MC for:

- a) Alexander (edge flip) moves
- b) Kawasaki moves (inclusion jump)

can be implemented. Nevertheless, for this, the tether constraints, and hard-core potentials (between every vertex) must be replaced with smooth (differentiable) potentials. This has not been proven so far to be sufficient to ensure self-avoidance (or might require inefficiently small-

time step). One could expect that hyper-flexible membranes exhibiting branching instability could lead to serious problems. Please note that hyper-flexible membranes can form (even at high bending rigidity) due to the presence of curvature-inducing inclusions (curvature instability, see also *Soft Matter* 15, 9974-9981 (2019)).

Moreover, a clear advantage of an MD-like algorithm is capturing the membrane dynamics, which without including a correct hydrodynamic model will not be realistic (see our response to question 8 of reviewer 3). Therefore, capturing dynamics should not be considered as an advantage of MD like algorithms for DTS systems.

Additionally, membranes exhibit shapes with very different configurational structures, separated by many large energy barriers. Therefore, MD-like algorithm may never be able to provide enough sampling while by MC or by creating some smarter MC moves (like parallel tempering implemented in FreeDTS or Hamiltonian replica exchange) this may be possible.

If the reviewer agrees, we would like to avoid adding any content to the manuscript regarding our response to this question as there has yet to be a rigorous evaluation of the advantages and disadvantages of MC and MD schemes for DTS models. Moreover, we are not strongly committed to MC schemes and if the problem in hand and future progresses demand MD-like algorithms, we will indeed move to that direction.

6) You mention that cytoskeleton can be added to the code. My feeling is that this would require parallel implementation or the use of accelerators, as the system size might become prohibitively large.

We agree with the referee, there is indeed a limit to system size without parallel implementation. However, cytoskeleton or other cytoplasmic materials and activities does not need to be directly implemented in the FreeDTS (or any other membrane simulation software). A separate software (such as MEDYAN) could be coupled to FreeDTS.

**Below paper was added to the manuscript regarding the MEDYAN software
Haoran Ni and Garegin A. Papoian *J. Phys. Chem. B* 2021, 125, 10710–10719; *Membrane-MEDYAN: Simulating Deformable Vesicles Containing Complex Cytoskeletal Networks***

Reviewer #3 (Remarks to the Author):

The authors study biomembranes at mesoscopic scales using a certain software called FreeDTS. The manuscript has a section on "The model" followed by a "Results" section. My main points of criticism are ordered in the same way:

We would like to clarify that this manuscript is not a study of biomembranes at mesoscale but rather provides and describes a platform (software that was not available before) to explore biomembranes at mesoscale. This distinction is important as several of the reviewer comments are rather about the study of biomembranes. We have indeed carefully considered all the points raised by the reviewer and have modified the manuscript accordingly and hope the modifications are fulfilling.

First about "The Model" section:

1) The abbreviation FreeDTS appears already in the title. In the abstract, it is stated that FreeDTS is some software. However, the authors do not explain the meaning of this abbreviation. Neither do they explain who developed this software.

We have chosen the name FreeDTS because it is free to use and free from any external library apart from the C++ Standard Library. The DTS part refer to dynamically triangulated surface.

We have added below text to the introduction to clarify the name use.

“We have chosen the name FreeDTS because it is free to use and free from any external library apart from the C++ Standard Library. DTS refers to Dynamically Triangulated Surfaces.”

The software is created by the authors and this paper attempt is to publish this software and make it available to the community. The first line of the abstract (repeated below in italic) indicates this. However, in the second sentence we change the work “model” to “software” to be clearer.

We present FreeDTS software for performing computational research on biomembranes at the mesoscale.

2) The method used here describes the lipid membranes as triangulated surfaces which are studied by Monte Carlo simulations. The vertices of the triangulation are decorated by “in-plane inclusions” representing membrane proteins. One confusing aspect about the simulation approach described here is the basic length scale used in this approach. On page 2, beginning of last paragraph, the edge length l_e of the triangles is introduced, which seems to provide the basic length scale of the approach. However, the length scale l_e is not mentioned in the figure captions of Figs. 2 - 6 where we find instead the length scale d_{dts} .

Thank you for pointing this out. The reviewer is correct, d_{dts} have been, by mistake, used for l_{dts} , which is the basic length unit of the model (and the software). In the new version of the manuscript, all the d_{dts} have been replaced by l_{dts} and l_{dts} has been defined in the early part of the model section that reads as.

To ensure self-avoidance, there is a hard-core potential between the vertices such that the minimum distance between any two vertices must be equal to l_{dts} (the basic length unit in FreeDTS). Additionally, self-avoidance requires that the edge length vary within a specific range ($l_{min} \leq l_e \leq l_{max}$). It has been tested that $l_{min} = l_{dts}$ and $l_{max} = \sqrt{3}l_{dts}$ with a mild constraint on the dihedral angle between two neighboring triangles is enough to ensure self-avoidance of the surface[35]. As a note, l_{min} , l_{max} and the minimum dihedral angle can be set by the user in FreeDTS.

For clarification, l_{dts} is constant, while l_e is not constant and can vary within the boundaries defined by the user.

3) It requires a systematic search through the whole main text to find out how d_{dts} has been defined. On page 6, first paragraph, one finds the statement that “the unit length is d_{dts} , which is the minimum distance between any pair of vertices”. However, it is completely unclear why the authors need the new notation d_{dts} because they introduced the minimum distance l_{min} of the edge length already on page 2, where they also set $l_{min} = l_{dts}$. As a result, the manuscript uses three different notations - l_{min} , l_{dts} , and d_{dts} - for the same basic length scale, which is very confusing and must be avoided.

This comment is related to previous one and it has been corrected now.

4) In addition to the notational confusion just described, the authors do not seem to make any attempt to estimate this length scale in terms of physical units, that is, in terms of nanometers. However, without such an estimate, it is hardly possible to assess the membrane conformations displayed in the figures of the “Results” section.

To address the reviewer’s comment (and comment 14), we have added several sections to the manuscript and the SI. The reasoning behind the changes and this specific presentation is as follows:

- 1) l_{dts} is basic unit length in FreeDTS. Therefore knowing (or fixing) l_{dts} in a physical unit, will fix any other length in the code and simulations. However, it will be unconstructive to fix this length within the code. This will limit its applications. This is**

actually very common in many simulation codes (a part software designed for all atom simulations as the logic is quite different). For example, in DPD specific software, the bead effective interaction radius is usually used for the basic unit length and only when a specific system is under study, this length is fixed (for example see: JC Shillcock, R Lipowsky The Journal of chemical physics 117 (10), 5048-5061). As matter of fact, it is rather more convenient to report data in a rescaled units and find general behaviors which can explain many different physical systems.

Therefore, l_{dts} remained unspecified within the code.

- 2) l_{dts} can be converted into a physical unit (such as nm) based on the constituent protein size. Such a conversion has been done in some of our previous works (Nature Communications 11, 2296 (2020), Soft Matter 17 308 (2021), Soft Matter 12 5164 (2016), and Front Mol Biosci 6 59 (2019)) that are cited in the manuscript. However, this approach is our preferred scheme while the software is not limited to this scheme, and users are free to design their ways therefore, we did not make this conversion.

We have added “Section 6: Converting l_{dts} to a physical unit” to the SI to make our scheme clear.

We also added the below sentence to the main manuscript to indicate why protein size can be used to fix l_{dts} .

“Note equation 6 (also equation 7) indicates that the effective interaction area of an inclusion with the membrane is A_u . Therefore, knowing the size of the protein will convert the l_{dts} to a physical unit such as nm [12, 38] (also see Section 6 of the SI).

- 3) The results presented in the manuscript Fig-3-6 are more generic compared to one single physical system and can be applied to much more. However, in the line to our answer to point 13 we convert these results to the case of Shiga and cholera toxins B subunits and added below paragraph in the “Membrane shape deformation by proteins” section to clarify this point.

“While these results can describe a wide range of processes involving remodeling of membrane shapes by proteins, one can convert l_{dts} to a physical unit when for example a specific protein is under consideration. For instance, if we consider that our proteins are B subunit of cholera or Shiga toxins (they have a similar lateral size of $\sim 7.2 \text{ nm}$ [38, 39]), then $l_{dts} \sim 6.9 \text{ nm}$ (see Section 6-SI). Therefore, $c_0 = 0.4 l^{-1} \sim 0.058 \text{ nm}^{-1}$. This is actually very close to the reported curvature induced by these proteins ($\sim 0.056 \text{ nm}^{-1}$ for cholera toxin and $\sim 0.07 \text{ nm}^{-1}$ for Shiga toxin). Also, in this case, the total surface area of the membranes will $\sim 0.15 \mu\text{m}^2$. For Shiga toxin, protein-protein interactions are primarily driven by close distance membrane fluctuation-induced forces $\sim 1 k_B T$, which cannot be captured by this model and must be included directly [50]. However, it is still unknown what causes the clustering of cholera toxin. Therefore, the results of the first and second columns of Figure 3 are expected for the B subunit of Shiga toxin which is also very similar to the shape reported in experimental settings [51]. In contrast, all configurations are possible for cholera toxin, depending on the range of its protein-protein interactions.”

- 4) The results presented in the manuscript is to show the power of FreeDTS in capturing a wide range of biomembrane systems. In every section we have provided reasoning on why any of this example are important.

5) In equation (1), the Gaussian curvature modulus is written with a minus sign which differs from the standard definition of this modulus.

We have defined the equation in this manner so that κ_G is positive, and positive $\Delta\kappa_G$ represent an increase in Gaussian modulus. We have added below sentence to avoid confusion when comparing to the literature.

“The third term is written with a minus sign so that the κ_G is positive. However, often in the literature a positive sign is used for the third term and therefore the reported value of κ_G is negative.”

6) The lateral size of the membrane proteins is not taken into account because these proteins are spatially confined to the vertices of the triangles, that is, they are described as point-like particles with some internal degrees of freedom corresponding to their orientation. Apparently, the authors assume here that the lateral size of the membrane proteins is small compared to the (average) edge length of the triangles but this assumption should be spelled out explicitly. Furthermore, this type of modeling becomes problematic when the lateral size of the proteins is large compared to the lateral size of the lipids, the latter being typically between 0.5 and 0.7 nm.

In response to this comment, we would like to drive the reviewer attention into two separate concepts of the model and the code.

- 1) In the numerical integration, vertices and inclusions are treated as a single particle.
- 2) In the physics of the model, a vertex has a character of a surface element (equation 1) and an inclusion is an object, interacting with the membrane. The area of the interaction is equal to the vertex area (equations 6 and 7).

Therefore, we do not assume that the protein size is smaller than the average edge length, rather we consider the effective protein size to be comparable to the edge size. This appears in the excluded volume effect (only one protein can occupy a vertex) and equations 6 and 7 that the energetic interaction of inclusion with the membrane is proportional to the area of the vertex. Therefore, for large proteins, the model works fine, and these proteins are the main target of mesoscopic modeling. The model, however, “requires some modification of the excluded volume contributions” when dealing with small proteins (smaller than membrane thickness, <4nm). In this case, a vertex could in principle own more than one inclusion, and therefore the excluded volume should be explicitly introduced into the system energy.

We have made this clear by adding below sentences to the main manuscript. See also our response to comment 4.

Equation 1 implicitly indicates that a vertex has the character of a surface element rather than a particle.

There is at most one inclusion per vertex, which naturally handles the in-plane excluded volume effect between inclusions.

Note equations 6 and 7 indicate that the effective interaction area of an inclusion with the membrane is A_v . Therefore, knowing the size of the protein will convert the l_{dts} to a physical unit such as nm. The model, however, requires some modification of the excluded volume contributions when dealing with small proteins (smaller than membrane thickness, <4nm). In this case, a vertex could in principle own more than one inclusion, and therefore the excluded volume should be explicitly introduced into the system energy and l_{dts} will be defined by the maximum number of inclusions occupying a vertex.

7) The interaction between two proteins on two neighboring vertices is described by equation (9). This interaction, which is claimed to represent “the simplest interaction between two neighboring inclusions”, depends on five model parameters. The numerical values chosen for these parameters are mentioned in some of the figure captions but it remains completely unclear where these numerical values come from.

We agree with the referee that the use of word “simplest” is rather misleading. We have changed this sentence to

The interaction between two neighboring inclusions i and j can be expressed as a Fourier expansion at the lowest order as

However, while the interaction energy is rather complicated (5 model parameters), not all the terms are always required. For instance, previously we had modeled Shiga toxin B subunit and Annexin 4 using only two terms (Pezeshkian et al, Soft matter 2016, Florentsen et al, Soft matter 2021). We have used this equation to provide fixability for the users to model different features of protein-protein interactions (available in the literature) as described in the manuscript.

We already had the below paragraph about how all model parameters should be obtained and what to do if not available. Also, more elaborated discussion on this has been provided in our perspective articles (Front Mol Biosci 6 59 (2019) and Biophysical Journal 122, 1883 (2023)).

To obtain biologically relevant information, models such as DTS may appear to be highly dependent on calibrating their parameters to start with. It should be noted, however, that this is not entirely accurate. Even without any knowledge of membrane shaping protein structure, DTS simulation can provide some knowledge about their structure by tuning the model parameters against macroscopic biophysical experiments [24, 58].

The result section of the manuscript is to examine the power of the software not to perform a full multiscale simulation of any specific biological system. Such a work is indeed well beyond the purpose of the current manuscript. However, the range of the parameters is chosen in the range of kT to represent weakly interacting proteins that many interesting phenomena emerges, and the exact reason why mesoscale is important (see for example Ruhoff, et al, Emerg Top Life Sci 7, 81-93 (2023)). For strongly interacting proteins, membrane configurations become the behaviors of macroscopic membrane with spontaneous curvature (rather well explored).

8) The Monte Carlo simulations described here do not take the aqueous solutions surrounding the membrane into account. Therefore, these simulations ignore the hydrodynamics of these solutions as well as the resulting hydrodynamic interactions between different membrane segments. These hydrodynamic interactions affect the relaxation of membrane undulations and protein orientations. Hydrodynamic interactions are also crucial during the time-dependent shape transformations from one membrane morphology to another. Therefore, the computational approach described here does not lead to a realistic dynamics of membranes and proteins.

Currently FreeDTS is designed to explore equilibrium shape of complex membranes. Equilibrium shape of complex (an even simple) membranes already is the answer to a wide range of important biological processes. Indeed, there are important membrane related processes that demands proper description of membrane dynamics and as the referee pointed out is out of the capacity of the current FreeDTS version. However, capturing hydrodynamics and dynamics are rather a difficult task and many even coarse-grained molecular dyanmcis simulations are not capable of doing it. Nevertheless, current framework used in FreeDTS allow to obtain in-plane properties and very well-suited to evaluate in-plane vector and tensor fields providing a great platform for coupling to hydrodynamic fields. We expect that in the future, FreeDTS will become capable of handling sufficiently accurate hydrodynamics and dynamics without significant computational costs.

To clarify this point, we have added the following text to the manuscript:

FreeDTS is currently designed to explore the equilibrium shape of complex membranes that is the answer to a wide range of key membrane-involved biological processes. Nevertheless, there are important biological processes that require a detailed description of membrane dynamics, consequently a correct membrane and solvent hydrodynamics, e.g., pearling instability [71], which is beyond the capabilities of the current FreeDTS version. Previously there have been some attempted to include the hydrodynamics effects, using implicit and explicit solvent particle, in both coarse grained and mesoscopic simulations [72-75]. Capturing realistic dynamics, in particular the effects of long-range hydrodynamics, is a challenging task and demands expensive computations. Nevertheless, the shape operator framework used in FreeDTS make it possible to evaluate in-plane vector and tensor fields, thereby providing a new strategy for coupling surface mechanics with hydrodynamics which in principle could have lower computational cost. Therefore, we expect that in the future, FreeDTS will become capable of handling sufficiently accurate hydrodynamics and dynamics without significant computational costs.

9) *The Monte Carlo approach has been previously extended to include hydrodynamics. One such extension is by Noguchi and Gompper in Phys. Rev. Lett. 93, 258102 (2004), which is based on a combination of multiparticle collision dynamics with Monte Carlo sampling.*

We are indeed aware of the work of Noguchi and Gompper. However, this method, with all its effectiveness, is associated with serious limitations (like many other methods). This has been discussed before, for example see Bolintineanu, et al, Comp. Part. Mech. (2014) 1:321–356 and Howard, Nikoubashman, and Palmer Current Opinion in Chemical Engineering 23 (2019): 34-43). Therefore, in our opinion, this method is not the first choice to capture the hydrodynamics effect in DTS. However, since it is a pioneering work of coupling the DTS method to hydrodynamics, we have added the paper to the list of references.

10) *Likewise, coarse-grained molecular dynamics simulations such as Dissipative Particle Dynamics (DPD) have been developed, which conserve momentum locally and provide a reliable description for the hydrodynamics of both membranes and aqueous solutions. Recent insights into the behavior of membranes and vesicles as obtained by DPD have been reviewed in Lipowsky et al, Biomolecules 13, 926 (2023).*

Yes, we agreed that for the explicit solvent method, DPD is rather an efficient method (see our answer to question 8). The paper mentioned by the referee and the below paper (among the pioneer works on applying DPD for membranes) was added to the list of references.

L. Gao, J. Shillcock, R. Lipowsky, Improved dissipative particle dynamics simulations of lipid bilayers, J Chem Phys 126 015101(2007).

11) *The authors emphasize the "frame tension" τ which is modeled by an additional energy term. This term is taken to be $-\tau A_p$, see page 3, first paragraph, where A_p represents the projected area. I am rather sceptical that the authors really want to include a minus sign here because positive values of τ would then lead to membrane compression rather than to membrane stretching.*

We have to disagree with the reviewer here. For positive τ , energy reduces by increasing A_p , therefore leads to stretching.

12) *In addition to the sign problem of the frame tension term, the magnitude of this tension is ill-defined when we consider the membranes of vesicles which represent the most popular membrane model systems. As a consequence, it is not clear how the authors derive the numerical values of τ as used in the manuscript, see, e.g., caption of Fig. 4.*

It is rather unclear what does reviewer mean by ill-defined. This tension (τ_{A_p}) is shown to be equal to the tension obtained from fluctuation spectrum which can be measured in GUVs. As supposed to tether formation in GUVs, Young–Laplace equation relates the pressure difference, induced for example by micropipette, to the mechanical tension.

As explain above, when considering a specific membrane system, l_{dts} can be converted to a physical unit and consequently the tension will have a physical unit. So the results, explains the behaviors of more than one single physical systems and can gives more specific values when one single one is considered.

Second, about the "Results" section:

13) Because of point 4 above, the size of the membrane morphologies displayed in Figs. 3 - 7 is unclear. The authors need to explain which experimentally accessible membranes are proposed to attain these morphologies on mesoscopic scales.

Please also see our response to point 2.

The results presented in the manuscript Fig-3-6 are more generic compared to one single physical system and can be applied to much more. However, in the line to our answer to point 4 we convert these results to the case of Shiga and cholera toxins B subunits and added below paragraph in the "Membrane shape deformation by proteins" section to clarify this point.

While these results can describe a wide range of processes involving remodeling of membrane shapes by proteins, one can convert l_{dts} to a physical unit when for example a specific protein is under consideration. For instance, if we consider that our proteins are B subunit of cholera or Shiga toxins (they have a similar lateral size of $\sim 7.2 \text{ nm}$ [38, 39]), then $l_{dts} \sim 6.9 \text{ nm}$ (see Section 6-SI). Therefore, $c_0 = 0.4 l^{-1} \sim 0.058 \text{ nm}^{-1}$. This is actually very close to the reported curvature induced by these proteins ($\sim 0.056 \text{ nm}^{-1}$ for cholera toxin and $\sim 0.07 \text{ nm}^{-1}$ for Shiga toxin). Also, in this case, the total surface area of the membranes will $\sim 0.15 \mu\text{m}^2$. For Shiga toxin, protein-protein interactions are primarily driven by close distance membrane fluctuation-induced forces $\sim 1 k_B T$, which cannot be captured by this model and must be included directly [50]. However, it is still unknown what causes the clustering of cholera toxin. Therefore, the results of the first and second columns of Figure 3 are expected for the B subunit of Shiga toxin which is also very similar to the shape reported in experimental settings[51]. In contrast, all configurations are possible for cholera toxin, depending on the range of its protein-protein interactions.

As a note: the purpose of Fig 7 was to demonstrate another capacity of the software (very useful for future development of multiscale simulations) and it was written in the legend that the proteins do not represent any realistic protein.

14) Mesoscopic scales can be probed by light microscopy. One very popular and highly useful membrane system is provided by giant unilamellar vesicles (GUVs). However, when observed by light microscopy, GUVs have smoothly curved membranes which look very different from the kinky shapes displayed in Figs. 3 - 7. This kinky appearance must be an artefact of the underlying triangulation of the membrane surfaces. Some snapshots appear to be relatively smooth. Did the authors apply some algorithm to smoothen these snapshots?

The results (snapshot) presented in the manuscript have not been processed (smoothed out) for visualization and they are direct output of the simulation.

We have to disagree with the referee on this point. While GUVs might be very smooth, GUV can exhibit very wild configurations when interacting with membrane proteins and even when they constitute some cone shape lipids such as GM1. For example, the below picture (Taken from Nature 450, 670–675 (2007)) shows how a GUV become very unsmooth due to the binding of the Shiga toxin B subunit. In the simulations, all the kinky shapes are the results of membrane interactions with inclusions.

Nature 450, 670–675 (2007)

15) Many of the shapes in Fig. 3 involve protein-rich and protein-poor membrane segments. These shapes represent examples for phase separation into two types of intramembrane domains which then undergo domain-induced budding. The latter process depends on the line tension of the domain boundary which can dominate the budding process as predicted theoretically in Lipowsky, J. Phys. II France 2, 1825 (1992) and observed by several experimental groups. The authors should explain why they chose to ignore this line tension.

In these simulations, line tension arising from the domain boundary is not ignored. The inclusion-inclusion interactions give rise to a line tension at a larger scale. While it is true that line tension could lead to budding, it is not the only driver for this process. For instance, the results show for small inclusion-inclusion interactions (and even zero interactions shown in Soft Matter 15(48) (2019) 9974-9981) budding can form.

We have added the following sentence to clarify this point in the manuscript and added the article mentioned by the referee to the references.

The non-zero protein-protein interactions lead to a line tension effect at the boundary that can energetically assist the budding process, especially for large protein-protein interactions[47]. However, budding may occur without a line tension effect solely as a result of the high concentration of inclusions that induce curvature [35].

A final and general comment:

16) The authors mention several alternative simulation approaches in the introduction of the manuscript but they do not explain why we need yet another simulation approach based on FreeDTS. In addition, they do not compare their results with the results of previous simulations. Such a comparison is necessary, however, in order to assess the advantages and disadvantages of the FreeDTS approach.

The reason for FreeDTS is that there is no freely available software for mesoscopic simulations using DTS approaches (with the exception of TriMem, which has a limited application and does not have a protein model). Other approaches are often case-specific and are still unavailable. So, the main advantage of FreeDTS is that it is available to everyone and allows exploring a wide range of processes. Moreover, it also contains several unique features that never have been developed for other methods e.g., constant tension algorithm, protein-membrane and protein-protein interactions and the example in Figure 7.

The initial version of the software has been used (by the authors and collaborators) to explore a wide range of biological processes (see for example ACS Nano 17, 966 (2022) and Soft Matter

17, 308 (2021)). Now, after more extension, the source code is being made open-source particularly due to the demand from the computational community. DTS schemes are an effective approach to exploring biomembrane shapes. But due to the inaccessibility of open-source software, its utilization has been a big challenge for new groups (huge amounts of work and time for something that is previously done). In contrast, for aaMD and cgMD users there are many different softwares. As such, not only FreeDTS is essential, but also more software of the same kind will be greatly appreciated by the community in order to push the boundaries of mesoscale membrane modeling.

The points made here, were already (in a compact version) in the manuscript in the below paragraph.

Several mesoscopic models have been used to explore diverse range of membrane associated processes such as protein clustering via membrane-mediated interaction [22], membrane shape remodeling by crowding of intrinsically disordered proteins[23], membrane neck constriction by assembly of proteins [24] and even activity-driven membrane remodeling [25] (for more see [20] and the reference within). In spite of this, these studies are often conducted with in-house software, or/and the software is limited to those specific applications that are difficult to apply to new research questions, which has hampered progress in mesoscopic membrane modeling.

Reviewers' comments:

Reviewer #1 (Remarks to the Author):

The authors have provided a substantially updated manuscript that will be an important contribution to the literature. The development of FreeDTS will be useful to a broad range of communities interested in mesoscale modeling of biological membranes. The ability to connect membrane morphology with protein inclusions, in particular, is likely to be an essential tool in addressing open questions about the regulation of membrane ultrastructure. While FreeDTS is not the final word in membrane modeling - note the absence of hydrodynamics, the simplistic form of protein-protein and protein-membrane interactions, and the absence of explicit lipid structure - it represents a robust compromise between utility and accuracy while providing a extensible framework that may, in time, be extended to more sophisticated descriptions of the microscopic mechanisms. I encourage the publication of this manuscript which introduces the field a useful new modeling tool for communities studying a broad range of different biological processes that depend on membrane ultra-structure: from cellular signaling to photosynthesis.

Reviewer #2 (Remarks to the Author):

The authors have adequately answered my specific questions. In response to more general concerns (code parallelisation, use of accelerators, implementation of membrane dynamics and hydrodynamics), the authors have confirmed them to be difficult, but potentially possible in the future. These would be the main limitations of the current software package.

As I mentioned before, this manuscript describes the freely available software, while scientific examples to illustrate the software are not a new scientific work. I was not sure whether the pure presentation of a software represents enough significance of this work.

Reviewer #3 (Remarks to the Author):

In their revised manuscript, the authors have addressed several of my concerns but did not understand some of my comments. As a consequence, the theoretical model implemented into their software has some severe flaws.

One serious problem is related to my previous point 11 which was:

11) The authors emphasize the "frame tension" τ which is modeled by an additional energy term. This term is taken to be $-\tau A_p$, see page 3, first paragraph, where A_p represents the projected area. I am rather sceptical that the authors really want to include a minus sign here because positive values of τ would then lead to membrane compression rather than to membrane stretching.

In their response, the authors say:

"We have to disagree with the reviewer here. For positive tau, energy reduces by increasing A_p , therefore leads to stretching."

In the following, I will explain my previous point 11 in more detail:

11a) The elastic energy is positive for both stretching and compression as follows from the stretching energy E_A in their Eqn (2), which is proportional to the area compressibility modulus K_A .

To be more precise, Eqn (2) is wrong in its present form because N_T is the number of triangles and A is the area of the triangulated surface as the author now explain by the red text piece in the 3rd line below Eqn (3). The stretching energy in Eqn (2) can be corrected by deleting the factor N_T .

After this correction, the tension in the membrane is obtained by taking the derivative of E_A with respect to A . This derivative leads to $K_A (A - A_0)/A_0$, which represents the mechanical tension Σ_m experienced by the membrane.

The authors should realize that the mechanical tension $\Sigma_m = K_A (A - A_0)/A_0$ is positive when $A > A_0$, that is, when the membrane is stretched.

11b) Digression on Eqn (2): after deleting the factor N_T , this equation represents the discrete version of the global expression for the stretching energy. In order to include area fluctuations, one should replace this expression by a sum over the triangles and consider the local deviations of the triangle areas from the average triangle area.

11c) In their manuscript, the authors consider two energy terms related to membrane tension. First, the stretching energy as given by the (corrected) Eqn (2) and, second, the "frame tension" term. Both tension terms involve the area of the triangulated surface. The stretching energy involves the true area A of the triangulated surface whereas the frame tension term is taken to be proportional to the "projected area" A_p of the triangulated surface. It is quite obvious that these two terms are intimately related to each other, but the authors erroneously assume that they represent two elastic energy terms which are independent of each other.

Thus, the theory described here involves some double-counting of the elastic stretching energy, which is a serious flaw of this theory.

11d) The bending energy in Eqn (1) of the manuscript represents a discrete variant of the continuum theory of curvature elasticity. In the latter theory, one usually adds the term ΣA to the bending energy of the membrane where Σ represents a Lagrange multiplier which is conjugate to the prescribed area A of the membrane. It turns out that the Lagrange multiplier tension Σ is, in fact, identical with the mechanical tension Σ_m , see Lipowsky, Adv. Colloid Interface Sci. 208: 14-24 (2014). This result corroborates my previous statements above: If the authors include the (corrected) Eqn (2), they should not include any additional "frame tension" term.

This brings me to my previous point 12 which was:

12) In addition to the sign problem of the frame tension term, the magnitude of this tension is ill-defined when we consider the membranes of vesicles which represent the most popular membrane model systems. As a consequence, it is not clear how the authors derive the numerical values of τ as used in the manuscript, see, e.g., caption of Fig. 4.

The response of the authors was:

"It is rather unclear what does reviewer mean by ill-defined. This tension (τ_{A_p}) is shown to be equal to the tension obtained from fluctuation spectrum which can be measured in GUVs. As supposed to tether formation in GUVs, Young–Laplace equation relates the pressure difference, induced for example by micropipette, to the mechanical tension."

My response to their response:

12a) I strongly disagree with this response. First, if they want to apply their "frame tension" term to a GUV, they need to define the projected area of this GUV. Because each GUV has a closed membrane surface, it does not possess a uniquely defined projected area. Second, I agree that the mechanical tension in the GUV membrane is typically deduced from the Young-Laplace equation but this tension is directly related to Eqn (2) without the factor τ , see my previous point 11b.

12b) Because the authors insist that they want to consider a frame tension term

τA_p , all examples in the figures of the manuscript correspond to membranes that are effectively COMPRESSED by positive values of τ . This compression explains why most shapes appear to be kinky as I noticed in my previous point 14.

12c) If the membrane is constrained by some "frame", it experiences some boundary conditions along this frame. Therefore, the "frame tension" term should be replaced by appropriate boundary conditions along the boundary of the membrane. Such boundary conditions are, of course, inappropriate, for GUVs and other vesicles which have no boundaries.

The authors intend to describe a freely available software code that can be applied to the mesoscale behavior of membranes. However, as explained above, the software described here is based on some serious misconceptions of the underlying theory. In addition, this theory is described in a rather sloppy, superficial, and disorganized manner, which will misguide possible users of the software.

After thoroughly reviewing all the comments provided by reviewer 3, we have reached the conclusion that the reviewer's strong disagreement stems from a misunderstanding of our algorithm and, subsequently, our previous responses. Possibly, the confusion arose because we were not entirely clear on what membrane in a periodic box is (membrane with periodic boundary condition), and why the possibility of box change is necessary. This has been clarified now in the manuscript. In this document, we first elaborate on the previous comment 11 and show that our response was indeed correct, and then provide point-by-point response to all the new points raised by the reviewer. We also provide substantial evidence confirming the robustness of our results and algorithm. We believe that the previous assessment of the reviewer was mostly due to unclarity in the description of some features of the algorithm and hope that the reviewer acknowledges this after reading this document.

For your reference: reviewer comment are inside gray boxes, our responses are under the corresponding box and specific important points are in bold. In the current version, we have added two new figures to the list of SI figures i.e., Figure SI-8.3 and Figure SI-8.4. This structured format aims to ensure clarity and comprehensive addressing of the reviewer's comments.

Reviewer #3 (Remarks to the Author):

In their revised manuscript, the authors have addressed several of my concerns but did not understand some of my comments. As a consequence, the theoretical model implemented into their software has some severe flaws.

One serious problem is related to my previous point 11 which was:

11) The authors emphasize the "frame tension" τ which is modeled by an additional energy term. This term is taken to be $-\tau A_p$, see page 3, first paragraph, where A_p represents the projected area. I am rather sceptical that the authors really want to include a minus sign here because positive values of τ would then lead to membrane compression rather than to membrane stretching.

In their response, the authors say:

"We have to disagree with the reviewer here. For positive tau, energy reduces by increasing A_p , therefore leads to stretching."

In the following, I will explain my previous point 11 in more detail:

Regarding frame tension term, i.e., equation $E_{A_p} = -\tau A_p$ that appears in the manuscript and related to previous comment 11.

Our previous response was indeed correct. Perhaps our previous response lacked sufficient detail. The algorithm was previously published (in 2019, Soft Matter 15, 9974-9981) and we had presumed

the reviewer was familiar with this prior work (we had cited this work in the related section in the manuscript). We acknowledge that our use of term "membrane in a periodic box" may not have been the best choice (although this term is commonly understood within the computational and simulation community) and using the term "membranes with periodic boundary conditions" might have been more precise. We have now changed this term in the manuscript and included a supplementary figure for visual clarification (note: the box in Figures 2 and 3 cannot be displayed due to visualization software limitations). Additionally, the physical basis behind membranes with periodic boundary conditions and the need for the box change is described in Appendix 1 at the end of the document.

We demonstrate the correctness of not only the equation's sign but also the algorithm itself through the presentation of four distinct types of evidence; i.e., 1) Literature and other expert publications; 2) Our simulation results confirm our claim regarding the sign of the equation; 3) The results of the algorithm correspond perfectly to what is reported in the literature (tension obtained from fluctuation spectrum); 4) Theoretically it is clear why the sign should be negative.

1) Literature and other expert publication: Energy term of $-\tau A_p$ has been used with the exact same format when describing thermodynamics behaviors of membranes. See for example:

The first term of Equation 3 and 13 in Hayato Shiba, Hiroshi Noguchia and Jean-Baptiste Fournier *Soft Matter*, 12, 2373-2380(2016).

In Equation 4 of Neder, Jörg, et al. "Coarse-grained simulations of membranes under tension."; *The Journal of chemical physics* 132 115101 (2010). ($-\Gamma A$, where is A projected area and Γ is the frame tension).

In Equation 19 of Durand, *Soft Matter*, 18, 3891-3901(2022) (L_p instead of A_p is used as the system is solved in one dimension in this paper).

In Equation 7 of Schmid, Friederike. "Are stress-free membranes really tensionless?." *Europhysics Letters* 95 28008 (2011).

2) Our simulation results confirm our claim regarding the sign of the equation. DTS simulations with positive value of τ (without any protein) yield a flat membrane for very large number of steps (10 million) see below Fig 1 (the red curve) and Fig 2 left side (also video `no_inclusion_t4.avi`). On the contrary, negative τ , compression and folding of the membrane surface occur rapidly (within a relatively small number of steps, 250 thousand), as illustrated by the blue curve in Figure 1 and the right side of Figure 2 (`no_inclusion_t_negative4.avi`).

Fig 1) Membrane projected area as a function of Monte Carlo steps (red) for $\tau = 4$, the projected area reaches equilibrium and then fluctuates. Inset (blue) $\tau = -4$; the projected area continuously decreasing, leading to membrane folding; see below snapshots.

Fig 2) Shape of the membrane for positive (left) and negative (right) values of τ .

3) With our algorithm we get exactly the tension obtained from fluctuation spectrum. As we also mentioned this in our previous response to the reviewer comments, this tension is equal to the tension obtained from undulation spectrum which is rather a common approach to obtain membrane tension and bending rigidity from both experimental systems and molecular simulations. In below Fig 3, we show this for 4 different values frame tension. As a note, this has been clarified and shown by

many other groups. For example, see: Durand, *Soft Matter*, 18, 3891-3901(2022) and Shiba et al, *Soft Matter*, 12, 2373-2380(2016).

Fig 3) Membrane tension obtained from undulation spectrum for different value of frame tension. Blue like is $f(x)=x$ showing that these two values are equal.

4) Theoretically it is clear why the sign should be negative. Consider a membrane within a periodic box in XY direction. The project area of the membrane (A_p) is equal to the box area in the XY plane. Now, with the assumption of $E = -\tau A_p$, for positive τ energy decreases as the A_p increases, meaning it stretches.

11a) The elastic energy is positive for both stretching and compression as follows from the stretching energy E_A in their Eqn (2), which is proportional to the area compressibility modulus K_A .

We would like to point out clearly that the $E = -\tau A_p$ equation (that was under question in comment 11), is different from equation 2. These two equations are different in nature, serve different function and modeling distinct effects. While equation 2 can be used (but not necessary) for every simulation, the frame tension term ($E = -\tau A_p$) is exclusively applicable for membranes with a periodic boundary condition (within a periodic box). If this distinction remains unclear, we kindly direct the reviewer attention to Appendix 1 at the end of this document.

For further clarification about the purpose of equation 2:

It appears that we and the reviewer agree on the fact that the solution to the membrane shape should be surfaces with constant area (real area not projected area) unless the system is coupled to a reservoir

of materials (grand canonical ensemble) or in high stretching regimes that some changes (rather small) can happen to the membrane area. The latter will be small as the membrane rupture happens in real membranes.

Inherently, in the framework of DTS simulation with our setup the area of the triangulated surface remains constant (with some fluctuations) as it is shown in the table SI-7.1. This is due to the hardcore potential between two neighboring vertices that only allow for edge sizes within a specific range (essential for self-avoidance). Indeed, we are not the first group to use this implementation, see for example: Biophys J. 5; 104 1018 (2013), Phys. Rev. E 99, 022414 (2019) and Phys. Rev. E 81, 041922 (2010). Therefore, in principle, one does not need to couple the system energy to equation 2 to keep the area constant. Also, it is recommended to not do so, since it reduces the sampling of the simulation. However, some other groups tend to apply such potential energy. As we wanted to create a platform (software) that provides all the possibilities, we have developed an algorithm for this. This was also very clear in our text. In other type of implementation, instead of a hardcore potential a tethering potential is applied on the edges. This potential also affects (in simulations) the stretching and compression that the reviewer is mentioning.

To be more precise, Eqn (2) is wrong in its present form because N_T is the number of triangles and A is the area of the triangulated surface as the author now explain by the red text piece in the 3rd line below Eqn (3). The stretching energy in Eqn (2) can be corrected by deleting the factor N_T .

The elastic stretching term Eq. (2) is an extensive energy variable, i.e., scale proportionally with the system size. Therefore, removing N_T will make this energy independent of the system size and thus not an acceptable energy form (in particular, for general purpose software). Also, the suggested derivation of $\frac{\partial E_A}{\partial A} = K_A \frac{(A-A_0)}{A_0}$ (according to the next point of the reviewer) indicates that the reviewer has the following form of equation 2 in mind, $E_A = \frac{K_A A_0}{2} \left(\frac{A-A_0}{A_0}\right)^2$ (not $E_A = \frac{K_A}{2} \left(\frac{A-A_0}{A_0}\right)^2$ that will be obtained by removing N_T from the Eq. (2)) where N_T is replaced by $A_0 = N_T a_T$. Here a_T is the reference total area per triangles. This form is also extensive and is equivalent to Eq. (2) up to a constant factor a_T (in the coupling constant).

Nevertheless, this equation has been presented in different forms in different articles (mostly due to convenience in the implementation). For example, in Lipowsky, Adv. Colloid Interface Sci. 208: 14-24 (2014), it is represented as

$$E_A = \frac{1}{2} K_A \frac{(A - A_0)^2}{A_0} \quad 11a.1$$

while in J. Chem. Phys. 157, 174801 (2022) it has been represented as

$$E_A = K_A \left(\frac{A - A_0}{A_0} \right)^2 \quad 11a.2$$

and in our manuscript as

$$E_A = K_A N_T \left(\frac{A - A_0}{A_0} \right)^2 \quad 11a.3$$

For a fixed simulation setup, the difference does not really matter because N_T and A_0 are constant numbers. This is rather a matter of opinion (and very common in scientific literature) on how to represent an equation as long as it is consistent in the rest of the formulation, which it is in our case. For example, Lennard Jones potential (a very well-known potential) are presented differently in GROMACS and CHARMM software packages (among the most widely used software's for molecular dynamics simulations).

After this correction, the tension in the membrane is obtained by taking the derivative of E_A with respect to A . This derivative leads to $K_A (A - A_0)/A_0$, which represents the mechanical tension Σ_m experienced by the membrane.

We do not have a “general” disagreement with what the reviewer has presented in this line, however, there are a minor error that should be considered as we state below.

Membrane tension (associated with total surface area) is the derivative of “free energy” with respect to area. Therefore, in addition to $K_A (A - A_0)/A_0$, there is another contribution to “this tension” that will arise due to the hard potential between the neighboring vertices or tether potentials in other implementation.

The authors should realize that the mechanical tension $\Sigma_m = K_A (A - A_0)/A_0$ is positive when $A > A_0$, that is, when the membrane is stretched.

Overall, we do not have much of a disagreement here with the reviewer. However, there appears to be some conflation between the term "compression" here and the subsequent discussion, particularly in comment 12b (related to $-\tau A_p$ and not equation 2). Here, "compression" signifies a reduction in the actual membrane area, which does not influence membrane folding or buckling. In contrast, the term "compression" in comment 12b, as well as the original comment 11, refers to a decrease in A_p area, leading to membrane buckling or budding.

11b) Digression on Eqn (2): after deleting the factor N_T , this equation represents the discrete version of the global expression for the stretching energy. In order to include area fluctuations, one

should replace this expression by a sum over the triangles and consider the local deviations of the triangle areas from the average triangle area.

If we have understood the reviewer correctly, the reviewer suggests a coupling energy as

$$E = \frac{k_a}{2} \sum_1^{N_T} \frac{(A_i - A_l)^2}{A_l} \quad (11.b - 1)$$

Just for clarity, $N_T A_l$, will be the reference triangulated area, while in equation 2, it is A_0 (reference triangulated area: the area at which the energy is minimum).

The reviewer does not mention the purpose of introducing such a term. So, we can only argue for why we have not included it in the Hamiltonian.

The Eqn(1) to Eqn(5) in the manuscript are terms kept in the spirit of Helfrich's Hamiltonian i.e. consisting only of the elementary surface invariants, involving area, mean curvature, Gaussian curvature, etc. Thus, in FreeDTS local variations in density do not give rise to energy changes as long as the total area is constant as one expects from a reparameterization invariant Hamiltonian. Already in 1976 de Gennes and co-workers considered an extension of the energy of type of equation 11:b-1 and concluded that internal density fluctuations decouple and becomes irrelevant for the equilibrium membrane conformation problem (see Brochard, F., P. G. De Gennes, and P. Pfeuty. "Surface tension and deformations of membrane structures: relation to two-dimensional phase transitions." Journal de Physique 37.10 (1976): 1099-1104).

11c) In their manuscript, the authors consider two energy terms related to membrane tension. First, the stretching energy as given by the (corrected) Eqn (2) and, second, the "frame tension" term. Both tension terms involve the area of the triangulated surface. The stretching energy involves the true area A of the triangulated surface whereas the frame tension term is taken to be proportional to the "projected area" A_p of the triangulated surface. It is quite obvious that these two terms are intimately related to each other, but the authors erroneously assume that they represent two elastic energy terms which are independent of each other.

Thus, the theory described here involves some double-counting of the elastic stretching energy, which is a serious flaw of this theory.

These two terms are not intimately related. We suspect that the reviewer's comment stems from a potential lack of clarity regarding our reference to a membrane in a periodic box (a membrane with periodic boundary conditions). If this is not the reason, we will provide further elaboration below to illustrate why these two terms are distinct.

The projected area of a membrane with a periodic boundary in the XY plane equals $A_p = L_x * L_y$ (L_x and L_y are the box side lengths in X and Y directions). For a constant A_p , the actual surface area (A)

can be any number larger than or equal to A_p . On the other hand, for a fixed A , A_p can be equal to or smaller than A . Since these two macroscopic variables are distinct, they require distinct couplings energy (For more information, see Appendix 1). Only for infinity stretched membranes these two areas are equal, which is physically impossible due to surface fluctuations (undulations).

There is a big body of literature on this topic, see for example:

- 1) Hayato Shiba, Hiroshi Noguchia and Jean-Baptiste Fournier Soft Matter, 2016, 12, 2373-2380(2016).
- 2) Durand, Soft Matter,18, 3891-3901(2022)
- 3) J.-B. Fournier and C. Barbetta, Phys. Rev. Lett., 100, 078103 (2008).

11d) The bending energy in Eqn (1) of the manuscript represents a discrete variant of the continuum theory of curvature elasticity. In the latter theory, one usually adds the term ΣA to the bending energy of the membrane where Σ represents a Lagrange multiplier which is conjugate to the prescribed area A of the membrane. It turns out that the Lagrange multiplier tension Σ is, in fact, identical with the mechanical tension Σ_m , see Lipowsky, Adv. Colloid Interface Sci. 208: 14-24 (2014). This result corroborates my previous statements above: If the authors include the (corrected) Eqn (2), they should not include any additional "frame tension" term.

We suspect that the reviewer is referring to the case of closed surfaces and not membrane segments in a periodic box. If yes, then we do not have much of disagreement here. As we pointed out above (and below in comment 12-a), the frame tension term is intended only for membranes with periodic boundary conditions.

Just for clarification:

In the numerical solution for membrane shape using the shape equation, $\Sigma_m A$, a Lagrange multiplier is used to ensure either

1) To keep the total area of the surface fixed:

In the framework of DTS, Equation 2, and Equation 11.b-1 (suggested by the reviewer and used in the mentioned paper) are ensuring this requirement. These equations form numerical approaches to keep the total surface area constant. This is the reason why the issues we raised regarding equation 11.b-1 may not be significantly problematic.

2) Σ_m is a chemical potential that allows for the flow of materials when it is connected to a reservoir.

In the framework of DTS, this ensemble cannot be achieved by either of Equation 2, and Equation 11.b-1. It requires an additional Algorithm. See (see Appendix 1) and the below paper.

Julian Weichsel and Phillip L. Geissler; PLOS Computational Biology (2016);

DOI:10.1371/journal.pcbi.1004982 July 6, 2016.

This brings me to my previous point 12 which was:

12) In addition to the sign problem of the frame tension term, the magnitude of this tension is ill-defined when we consider the membranes of vesicles which represent the most popular membrane model systems. As a consequence, it is not clear how the authors derive the numerical values of τ as used in the manuscript, see, e.g., caption of Fig. 4.

The response of the authors was:

"It is rather unclear what does reviewer mean by ill-defined. This tension (τ_{A_p}) is shown to be equal to the tension obtained from fluctuation spectrum which can be measured in GUVs. As supposed to tether formation in GUVs, Young–Laplace equation relates the pressure difference, induced for example by micropipette, to the mechanical tension."

My response to their response:

12a) I strongly disagree with this response. First, if they want to apply their "frame tension" term to a GUV, they need to define the projected area of this GUV. Because each GUV has a closed membrane surface, it does not possess a uniquely defined projected area. Second, I agree that the mechanical tension in the GUV membrane is typically deduced from the Young-Laplace equation but this tension is directly related to Eqn (2) without the factor N_T , see my previous point 11b.

As clarified earlier, the frame tension algorithm is specifically intended for membranes with periodic boundary conditions. So, we have not applied the frame tension algorithm to closed systems such as vesicles. Additionally, it's important to note that there is no mention of frame tension in the manuscript sections discussing Figures 5, 6, and 7.

In our initial response, we assumed that the concept of membranes within a periodic box was clear to the reviewer (and is clear that we have not applied this algorithm for GUV simulations). Thus, we presumed the reviewer question is about the meaning of τ when a segment of a GUV membrane is considered.

Nevertheless, in our previous response, we had only two statements that none of them are really incorrect.

- 1) First, we stated, "This tension (τ_{A_p}) is shown to be equal to the tension obtained from the fluctuation spectrum which can be measured in GUVs"
Based on the results we showed above (Fig 3) which are the same results as Hayato Shiba, Hiroshi Noguchia and Jean-Baptiste Fournier Soft Matter, 2016, 12, 2373-2380(2016). this statement is correct.

Also, recent work shows that the fluctuation tension is indeed equal to frame tension. Durand, Soft Matter, 18, 3891-3901(2022).

- 2) In the second point we stated, "As supposed to tether formation in GUVs, Young–Laplace equation relates the pressure difference, induced for example by micropipette, to the mechanical tension." In the new comment, the reviewer is acknowledging that this statement is correct.

12b) Because the authors insist that they want to consider a frame tension term $-\tau A_p$, all examples in the figures of the manuscript correspond to membranes that are effectively COMPRESSED by positive values of τ . This compression explains why most shapes appear to be kinky as I noticed in my previous point 14.

As demonstrated earlier, the sign in the equation $-\tau A_p$ is indeed accurate and the membranes are not compressed. We hope that the reviewer acknowledges this now. However, to bolster the reviewer's confidence in our findings, we present a series of supporting points.

- 1) Buds and tubulated shapes shown in the manuscript are produced by proteins within the system. We performed the same simulation, but without proteins. Two videos, no_inclusion_t0.avi and no_inclusion_t4 (attached with our files) show the trajectories of these simulations for 10 million steps. One is for $\tau=0$ and other is for $\tau=4$. It is clear that the membranes are not compressed. Please note, this is the same as the red line in Fig 1 above).
- 2) video no_inclusion_t_negative4.avi, shows the trajectory for a simulation with $\tau=-4$. It can be seen (only if we follow the reviewer's suggestion) the membrane is compressed (rather within a small number of steps).
- 3) Figure 3 in the main manuscript are simulations for $\tau=0$. Therefore, the sign never plays any role in these simulations.

We hope that the reviewer is convinced that these membranes are not compressed.

12c) If the membrane is constrained by some "frame", it experiences some boundary conditions along this frame. Therefore, the "frame tension" term should be replaced by appropriate boundary conditions along the boundary of the membrane. Such boundary conditions are, of course, inappropriate, for GUVs and other vesicles which have no boundaries.

We do not have any disagreement with the reviewer here, and neither this statement has any contradictions to our work.

- 1) As we have said above, the frame tension algorithm has not (and should not) been applied to any closed surfaces such as GUVs.

- 2) For periodic membranes (membrane with periodic boundary condition), the periodicity is the boundary condition that the reviewer is suggesting, and the frame tension algorithm guarantees a correct thermodynamics ensemble both for the variable A_p , i.e., (N, τ, T) and fixed A_p i.e., (N, A_p, T) .

The authors intend to describe a freely available software code that can be applied to the mesoscale behavior of membranes. However, as explained above, the software described here is based on some serious misconceptions of the underlying theory. In addition, this theory is described in a rather sloppy, superficial, and disorganized manner, which will misguide possible users of the software.

We believe that we have provided substantial evidence confirming the robustness of our results and algorithm. We believe that the previous assessment was due to unclarity of some features of the algorithm and hope that the reviewer acknowledges this now.

Appendix 1: Membranes with periodic boundary conditions.

A membrane in the periodic box represents a segment of a full closed membrane. To elaborate, it could signify a smaller portion of a Giant Unilamellar Vesicle (GUV), where its size is considerably smaller than the GUV radius, allowing it to be approximated as flat. Similarly, it might denote a segment of cellular membranes. These membrane segments are connected on the opposing side through the periodic box and are frequently employed in particle-based simulations like molecular dynamics. Given that membranes undergo limited stretching, two options are available to accurately capture membrane deformation and the associated thermodynamics of membrane shape:

- 1) To change the surface area of the membrane inside the box (**a grand canonical ensemble with a chemical potential as a coupling constant**). See for example: Julian Weichsel and Phillip L. Geissler; PLOS Computational Biology (2016); DOI:10.1371/journal.pcbi.1004982 July 6, 2016.
- 2) **Dynamic box size with constant frame tension (N, τ, T) ensemble. In this case, the box can change in the XY plane (the membrane plane)**. In this case, the energy term $-\tau A_p$ determines the energy penalty for the changes in the box in XY plane. Please note, in this case, A_p is the same as the size of the box in the XY plane. This ensemble is very common in particle-based simulations of membranes (see for example Reynwar et al, Nature 447, 461–464 (2007)). This approach, closely resembles molecular dynamics and Monte Carlo simulations of molecular systems with explicit solvent, utilizing the "constant pressure simulation algorithm" (often referred to as NPT ensemble). In these simulations, the pressure is coupled to volume, which is a standard setup of many simulations, particularly membrane simulations. Please note, in membrane simulations, the system is coupled to semi-isotropic pressure coupling, which couples the projected area of the membrane. (Understanding Molecular Simulation; From Algorithms to Applications By Daan Frenkel, Berend Smit Hardback ISBN: 9780122673511 chapter 5 section 4).

We also have taken the second strategy as it is more numerically sound, and the number of computations does not change in different stages of system evolution. Therefore, the purpose of τA_p is not to allow for changes in the real surface area of membranes, but to allow for the box change, in another word, change in the projected area.

REVIEWER COMMENTS

Reviewer #1 (Remarks to the Author):

The manuscript has been substantially clarified compared to the original version and additional calculations have been included to further clarify the algorithm. This manuscript now reads smoothly and provides a concise but thorough description of the different aspects of the FreeDTS software.

I have read through the author's extended response to a previous reviewer's comments. I'm convinced that the sign of the tension term is, indeed, correct (and consistent with several other publications from different groups in the literature), that it is appropriate to have both a frame tension and a stretching energy as distinct terms (the discussion by Durand 2022 that the author's cited was informative), and that the frame tension was (correctly) not applied to the simulations of closed vesicles. Having read through the reviewer's second round of comments (I could not find any copy of the first round of comments) and the author's response to those comments, I do not see any evidence of substantial technical errors in this manuscript.

I remain convinced that this manuscript represents an important contribution to the literature where the connection between protein density and membrane ultra-structure is becoming pressing in several related fields due to the rapid development of in situ cryo-electron tomography measurements. While FreeDTS in its current form is certainly not the final answer to such questions, the manuscript reports the development of an open-source code with the right foundations for further extensions that make it an attractive tool for addressing these problems in the future.

Reviewer #3 (Remarks to the Author):

In their long rebuttal, the authors try to argue that their treatment of membrane tension is meaningful. However, their arguments do not address my main point of criticism which was that "the theory described here involves some double-counting of the elastic stretching energy, which is a serious flaw of this theory".

I will now describe this double-counting once more, using a simpler and shorter line of arguments:

1) In their rebuttal, the authors agree that their simulations are based on a simulation box with periodic boundary conditions. The simulation box has a certain base area which is equal to the projected area A_p of the membrane. Thus, the simulation box plus boundary conditions impose a certain projected area A_p on the membrane.

2) In addition, their equation (2) provides an energy term E_A , which acts to impose the preferred area A_0 on the membrane.

3) Together, the projected area A_p imposed by the simulation box and the energy term E_A with the preferred area A_0 generate a tension in the membrane as soon as A_p differs from A_0 .

It is important to note that this membrane tension arises from the mismatch between A_p and A_0 and not from any additional energy term such as $-\tau A_p$ as used by the authors. Therefore, when the authors add this latter term to their energy, they double-count the elastic energy of the membrane.

4) For real membranes, one has to distinguish two tension regimes: an initial regime dominated by shape fluctuations of the membrane, in which the tension pulls out excess area from these fluctuations; and a second regime, in which the membrane is essentially flat and the tension stretches the membrane on molecular scales.

It is clear that the authors will have difficulties to accept the above line of arguments because they already used their theory or "model" in previous studies, emphasizing the "frame tension" τ as a crucial concept in all of these studies. Nevertheless, I urge the authors to think carefully about the above points 1) to 4); I am confident that they will eventually realize the correctness of these points.

Reviewer #4 (Remarks to the Author):

This is an interesting paper that describes a mesoscopic simulation methodology that appears to be able to simulate membranes undergoing a number of processes, including large scale remodeling events driven by proteins. Overall it looks sound to me. My main concern is one of novelty, or perhaps in trying to be more generous, one of proper scholarship. As far as I know, the first time this kind of approach was developed and applied (a discretization of the Helfrich, etc Hamiltonian) was in these two (not cited) papers written some time ago:

G. S. Ayton, P. D. Blood, and G. A. Voth, "Membrane Remodeling from N-BAR Domain Interactions: Insights from Multiscale Simulation," *Biophys. J.* 92, 3595-3602 (2007).

G. S. Ayton, R. D. Swenson, C. Mim, V. Unger, and G. A. Voth, "New Insights into BAR Domain Induced Membrane Remodeling", *Biophys. J.* 97, 1616–1625 (2009).

I see many of the same ideas from those papers in this present paper. These should be cited up front at the bottom of page 1 and credit should be given as them being the first for this kind of approach.

Then, and this is quite important for the authors in demonstrating novelty of their work, they should discuss at length how their work relates to (and if it improves upon) the methods described in this (not cited) paper:

A. Davtyan, M. Simunovic, and G. A. Voth, "The Mesoscopic Membrane with Proteins Model (MesM-P)", *J. Chem. Phys.* 147, 044101 (2017).

Without seeing these modifications of the text I am at present unable to judge the novelty of this work relative to others.

We would like to thank the reviewers for their thorough reading of our manuscript and response letter. We have carefully considered the points raised by reviewer 4 and modified the manuscript accordingly. We hope that our responses are fulfilling. We have also provided a point-by-point response to the reviewer 3 new comments. The referee comments are in italic and our responses are in bold. All the modified texts in the main manuscript are highlighted in red.

Reviewer #1 (Remarks to the Author):

The manuscript has been substantially clarified compared to the original version and additional calculations have been included to further clarify the algorithm. This manuscript now reads smoothly and provides a concise but thorough description of the different aspects of the FreeDTS software.

I have read through the author's extended response to a previous reviewer's comments. I'm convinced that the sign of the tension term is, indeed, correct (and consistent with several other publications from different groups in the literature), that it is appropriate to have both a frame tension and a stretching energy as distinct terms (the discussion by Durand 2022 that the author's cited was informative), and that the frame tension was (correctly) not applied to the simulations of closed vesicles. Having read through the reviewer's second round of comments (I could not find any copy of the first round of comments) and the author's response to those comments, I do not see any evidence of substantial technical errors in this manuscript.

I remain convinced that this manuscript represents an important contribution to the literature where the connection between protein density and membrane ultra-structure is becoming pressing in several related fields due to the rapid development of in situ cryo-electron tomography measurements. While FreeDTS in its current form is certainly not the final answer to such questions, the manuscript reports the development of an open-source code with the right foundations for further extensions that make it an attractive tool for addressing these problems in the future.

We thank the reviewer for conducting a thorough review of our response letter. We genuinely appreciate the positive assessment of our manuscript and the validation of the accuracy of our algorithms and methodologies.

Reviewer #3 (Remarks to the Author):

In their long rebuttal, the authors try to argue that their treatment of membrane tension is meaningful. However, their arguments do not address my main point of criticism which was that "the theory described here involves some double-counting of the elastic stretching energy, which is a serious flaw of this theory".

I will now describe this double-counting once more, using a simpler and shorter line of arguments:

It's disheartening to note that despite our comprehensive response, which included extensive information and state-of-the-art results drawn from the literature, the reviewer still maintains reservations. Before delving into a point-by-point answer to the reviewer's new comments, we wish to once again draw their attention to Figure SI-8.4. In this figure, our results directly and clearly show a perfect matching between the frame tension input and the fluctuation tension output that also has been shown in theoretical works (that we provided in our previous response). Therefore, if there were any instances of double counting, we would have observed inconsistencies in this particular result.

1) In their rebuttal, the authors agree that their simulations are based on a simulation box with periodic boundary conditions. The simulation box has a certain base area which is equal to the projected area A_p of the membrane. Thus, the simulation box plus boundary conditions impose a certain projected area A_p on the membrane.

In general, we do not have a major disagreement with the reviewer on this statement.

2) In addition, their equation (2) provides an energy term E_A , which acts to impose the preferred area A_0 on the membrane.

Regarding the discussion below, we generally concur with this statement. However, it's important to note that this is not the complete story, as we have elaborated on this in our previous response. We kindly direct the reviewer to our previous response in the "For further clarification about the purpose of equation 2" section for a more in-depth explanation.

3) Together, the projected area A_p imposed by the simulation box and the energy term E_A with the preferred area A_0 generate a tension in the membrane as soon as A_p differs from A_0 . It is important to note that this membrane tension arises from the mismatch between A_p and A_0 and not from any additional energy term such as $-\tau A_p$ as used by the authors. Therefore, when the authors add this latter term to their energy, they double-count the elastic energy of the membrane.

We suspect that the difficulty in reaching a common consensus rely on a different view of the Statistical Mechanics of this system, therefore below we obtain $-\tau A_p$ term using purely statistical mechanical approach.

We can start from Helmholtz free energy, $F(A, A_p, A_0, \dots)$; where A_0 is the number of the molecule multiplied by the area per molecule (different representation of particle number). Please note that the three dots stand for other macroscopic variables such as temperature and bending rigidity and not τ and σ . From Helmholtz free energy, the tensions can in principle be determined as

$$\tau = \left(\frac{\partial F}{\partial A_p} \right)_{A_0, A, \dots} \quad \text{and} \quad \sigma = \left(\frac{\partial F}{\partial A} \right)_{A_0, A_p, \dots}$$

Here, both tensions are dependent on model parameters and the constraints. For an ensemble that τ is the fixed (with variable A_p), the relevant Legendre transformed free energy (here denoted as R) will be

$$R(A, \tau, A_0, \dots) = F(A, A_p, A_0, \dots) - \tau A_p$$

And σ can be obtained as

$$\sigma = \left(\frac{\partial R}{\partial A} \right)_{A_0, \tau, \dots}$$

In this situation, the added term $-\tau A_p$ is important, as mentioned in the manuscript and previous response, to ensure that the MC sampling takes place at constant τ and variable A_p .

To clarify further:

We assume that in the reviewer statement, E_A refers to Equation 2. It's important to note that Equation 2 leads to energy change when A differs from A_0 , not A_p . Therefore, even when the total membrane area remains constant (for instance, when it equals A_0), there can still be numerous configurations with varying A_p . This implies that the energy associated with Equation 2 may remain unaltered even when A_p deviates from A_0 .

Please note, the frame tension represents a mechanical tension exerted on the membrane. In contrast, Equation 2, in addition to the tether potential, models effective molecular forces that fixes the total membrane area.

4) For real membranes, one has to distinguish two tension regimes: an initial regime dominated by shape fluctuations of the membrane, in which the tension pulls out excess area from these fluctuations; and a second regime, in which the membrane is essentially flat and the tension stretches the membrane on molecular scales.

We generally do not have a disagreement with the statement of the reviewer. As a matter of fact, in our previous response we stated this (provided below in blue italics). For membranes with fixed number of lipids, there are two regimes of membrane tension. A low-tension entropic elasticity regime and a high-tension elastic stretching regime, which terminates at the high lysis tension. The same behavior is found for the triangulated surface rigid membrane model except the stretching regime is not terminated by lysis at high tension but by the inextensible tether potential (unless it is coupled to another algorithm that allows for edge break). Exactly as our results show, in the lower limit, the roughness of the membrane shape due to membrane fluctuations is flattened by membrane tension. In the second regime, membrane stretching occurs and the membrane tension will compete with the term in Equation 2.

It appears that we and the reviewer agree on the fact that the solution to the membrane shape should be surfaces with constant area (real area not projected area) unless the system is coupled to a reservoir of materials (grand canonical ensemble) or in high stretching regimes that some changes (rather small) can happen to the membrane area.

It is clear that the authors will have difficulties to accept the above line of arguments because they already used their theory or "model" in previous studies, emphasizing the "frame tension" τ as a crucial concept in all of these studies. Nevertheless, I urge the authors to think carefully about the above points 1) to 4); I am confident that they will eventually realize the correctness of these points.

Our model is constructed upon a foundation of prior theoretical and computational research. We previously provided many lines of reasoning, with many literature references from multiple independent groups. Therefore, our confidence in the model is rooted in their robustness and alignment with previous studies, rather than any bias stemming from our own prior work. Indeed, should any evidence emerge demonstrating our model's inaccuracies, we will embrace it without any hesitations and adjust our model accordingly. Nonetheless, all indications currently support the solidity of our model.

Reviewer #4 (Remarks to the Author):

This is an interesting paper that describes a mesoscopic simulation methodology that appears to be able to simulate membranes undergoing a number of processes, including large scale remodeling events driven by proteins. Overall it looks sound to me. My main concern is one of novelty, or perhaps in trying to be more generous, one of proper scholarship. As far as I know, the first time this kind of approach was developed and applied (a discretization of the Helfrich, etc Hamiltonian) was in these two (not cited) papers written some time ago:

G. S. Ayton, P. D. Blood, and G. A. Voth, "Membrane Remodeling from N-BAR Domain Interactions: Insights from Multiscale Simulation," Biophys. J. 92, 3595-3602 (2007).

G. S. Ayton, R. D. Swenson, C. Mim, V. Unger, and G. A. Voth, "New Insights into BAR Domain Induced Membrane Remodeling", Biophys. J. 97, 1616–1625 (2009).

I see many of the same ideas from those papers in this present paper. These should be cited up front at the bottom of page 1 and credit should be given as them being the first for this kind of approach.

We thank the reviewer for having a positive opinion of our manuscript. We regret to have missed citing these articles. We now have added the papers to the list of the references.

Then, and this is quite important for the authors in demonstrating novelty of their work, they should discuss at length how their work relates to (and if it improves upon) the methods described in this (not cited) paper:

A. Davtyan, M. Simunovic, and G. A. Voth, "The Mesoscopic Membrane with Proteins Model (MesM-P)", J. Chem. Phys. 147, 044101 (2017).

Thanks for the very nice suggestion. First, to clarify, FreeDTS in addition to a model, is also a software that allows for running DTS simulations. As to the mesoscopic model described in the manuscript: While there are some similarities between our model and the aforementioned model, there are important fundamental differences. To clarify this, we have added the below paragraph to the discussion section of the main manuscript.

In a series of pioneering works, Voth and coworkers introduced a mesoscopic membrane model (called EM2, later upgraded to MesM-P) [23, 24, 68] that can be performed using LAMMPS open-source molecular dynamics package[69, 70]. This model has been successfully used to describe morphological changes of membranes by BAR-domain proteins. While described model in this manuscript shares certain similarities with MesM-P, there exist fundamental distinctions. Firstly, in FreeDTS, a membrane is explicitly represented as a surface, and the evolution of this surface is governed by the simultaneous adherence to self-avoidance principles and the preservation of the manifold configuration of the surface. In contrast, in MesM-P, the starting point is a coarse-grained model in fluid dynamics, where the explicit solvent and membrane components are represented by quasi-particles and the membrane's bending energy is included through a particle position with respect to its nearest neighbor membrane particles. The explicit representation of the surface becomes important at least in highly folded membranes where the nearest neighbors of a particle on a 3-dimensional space differ from the one on the membrane surface. Such morphologies are notably prevalent in the structural configuration of subcellular membranes, exhibiting high topological genus. Furthermore, the explicit membrane surface in FreeDTS allows for correct measurement of quantities such as system volume, surface area, and surface topology and offers several advantages, such as the utilization of numerous algorithms and geometrical descriptions originally developed for image processing techniques providing a wealth of algorithms to adopt for mesoscopic modeling of membranes. Secondly, FreeDTS allows for a more accurate representation of proteins. It permits anisotropic interactions between membrane proteins, making it especially suitable for elongated proteins. Additionally, it enables the modeling of proteins that induce changes in the membrane's Gaussian modulus, a factor that has been demonstrated to be critical for the emergence of membrane-mediated interactions. Last but not least, FreeDTS allows for parallel transport which is very important for proper modeling of protein-protein interactions on curved surfaces.

Without seeing these modifications of the text I am at present unable to judge the novelty of this work relative to others.

We now have added the discussion requested by the reviewer and hope that the modification is fulfilling.

REVIEWERS' COMMENTS

Reviewer #4 (Remarks to the Author):

I am generally happy with the revisions made by the authors, especially in terms of pointing out the similarities and differences of their model with the MesM-P (formerly EM2) model. However, I think two clarifications are in order. The authors go to considerable length in their added text to describe what they view as the superiority of their approach for describing membranes over the EMesM-P quasiparticle-based approach. However, it seems to me that they misrepresent the advantages (or disadvantages as they see it) of the quasiparticle approach of MesM-P. Unless I'm mistaken, the quasiparticle approach can handle to a certain degree the highly curved membranes through the anisotropic nature of the quasiparticle interactions and this could be mentioned, even if it's not as robust as the authors' approach to a "membrane sheet" in their eyes. Secondly and more importantly, because the quasiparticles in the MesM-P approach have potentials capable of dissociation, MesM-P can handle cases such as vesiculation or tubulation/fragmentation of membranes as driven by proteins like BAR domain proteins, as seen in the earlier Ayton et al work cited by these authors. Unless I'm mistaken, I don't think the authors' approach can handle such dramatic protein-driven membrane remodeling events that break into pieces?

Some clarification on these two points in the text would be a valuable addition, unless I'm wrong about them.

Reviewer #4 (Remarks to the Author)

I am generally happy with the revisions made by the authors, especially in terms of pointing out the similarities and differences of their model with the MesM-P (formerly EM2) model. However, I think two clarifications are in order. The authors go to considerable length in their added text to describe what they view as the superiority of their approach for describing membranes over the EMesM-P quasiparticle-based approach. However, it seems to me that they misrepresent the advantages (or disadvantages as they see it) of the quasiparticle approach of MesM-P. Unless I'm mistaken, the quasiparticle approach can handle to a certain degree the highly curved membranes through the anisotropic nature of the quasiparticle interactions and this could be mentioned, even if it's not as robust as the authors' approach to a "membrane sheet" in their eyes. Secondly and more importantly, because the quasiparticles in the MesM-P approach have potentials capable of dissociation, MesM-P can handle cases such as vesiculation or tubulation/fragmentation of membranes as driven by proteins like BAR domain proteins, as seen in the earlier Ayton et al work cited by these authors. Unless I'm mistaken, I don't think the authors' approach can handle such dramatic protein-driven membrane remodeling events that break into pieces?

Some clarification on these two points in the text would be a valuable addition, unless I'm wrong about them.

Our intent was never to misrepresent the MeshM-P work. We've revised the section in response to the reviewer's request, hoping it adequately addresses their concerns.

Regarding point one made by the reviewer: Previously, we explicitly mentioned that MeshM-P has been successfully used describe morphological changes of membranes by BAR-domain proteins. To refine this, we've revised the sentence as follows

“This model has been successfully used to describe complex membrane morphologies induced by BAR-domain proteins.”

Regarding the reviewer's second point: We completely agree. In our earlier manuscript, we explicitly stated that we hadn't incorporated a topology change algorithm yet, earmarking it for future implementations. Consequently, we hesitated to compare the forthcoming FreeDTS algorithm with MeshM-P's capabilities. To address the reviewer's suggestions, we've modified the paragraph to highlight this MeshM-P's capacity and the similar potential anticipated for FreeDTS. The full paragraph read as below.

In a series of pioneering works, Voth and coworkers introduced a mesoscopic membrane model (called EM2, later upgraded to MesM-P) [23, 24, 68] that can be performed using LAMMPS open-source molecular dynamics package[69, 70]. This model has been successfully used to describe complex membrane morphologies induced by BAR-domain proteins. While the model described in this manuscript shares certain similarities with MesM-P, there exist fundamental distinctions. Firstly, in FreeDTS, a membrane is explicitly represented as a surface, and the evolution of this surface is governed by the simultaneous adherence to self-avoidance principles and the preservation of the manifold configuration of the surface. In contrast, in MesM-P, the starting point is a coarse-grained model in fluid dynamics, where the explicit solvent and membrane components are represented by quasi-particles and the membrane's bending energy is included through a particle position with respect to its nearest neighbor membrane particles. Effects of curvature and topology changes e.g., membrane fragmentation, are described through anisotropic quasi-particle interactions, while they are handled with geometrical quantifiers in FreeDTS. Note, the current version of FreeDTS does not allow for surface topology change, however, it can be achieved by the

addition of certain discontinuous Monte Carlo moves (see below)[71, 72]. The explicit representation of the membrane surface in FreeDTS is important, in particular, in the modeling of highly curved and folded membranes, e.g., subcellular membranes with tubular or high-genus structures (see Figure 5-D). It also allows for correct measurement of quantities such as system volume, surface area, local curvature, and surface topology and offers several advantages, such as the utilization of numerous algorithms and geometrical descriptions originally developed for image processing techniques providing a wealth of algorithms to adopt for mesoscopic modeling of membranes. Also, FreeDTS allows for a more accurate representation of membrane proteins coupling to membrane curvature making it especially suitable for elongated proteins. Additionally, it enables the modeling of proteins that induce changes in the membrane's Gaussian modulus, a factor that has been demonstrated to be critical for the emergence of membrane-mediated interactions. Last but not least, FreeDTS allows for parallel transport which is very important for proper modeling of anisotropic in-plane interactions between membrane proteins on curved surfaces.